# Targeted inhibition of activated protein C by a non-active-site inhibitory antibody to treat hemophilia

Xiao-Yan Zhao[1✉], Andreas Wilmen[2], Dongli Wang[3], Xinquan Wang [3], Maxine Bauzon[1], Ji-Yun Kim[1], Lars Linden[2], Liang Li[3], Ursula Egner[4], Tobias Marquardt[4], Dieter Moosmayer[4], Jan Tebbe[2], Julian Marius Glück[2], Philipp Ellinger[2], Kirk McLean[1], Shujun Yuan[1], Subramanian Yegneswaran[1], Xiaoqiao Jiang[1], Vince Evans[1], Jian-Ming Gu[1], Doug Schneider[1], Ying Zhu[1], Yifan Xu[1], Cornell Mallari[1], Ashley Hesslein[5], Yan Wang[1], Nicole Schmidt[1], Katrin Gutberlet[6], Christine Ruehl-Fehlert[6], Alexius Freyberger[6], Terry Hermiston[1], Chandra Patel[1], Derek Sim[1], Laurent O. Mosnier [7✉] & Volker Laux[8✉]

Activated protein C (APC) is a plasma serine protease with antithrombotic and cytoprotective functions. Based on the hypothesis that specific inhibition of APC's anticoagulant but not its cytoprotective activity can be beneficial for hemophilia therapy, 2 types of inhibitory monoclonal antibodies (mAbs) are tested: A type I active-site binding mAb and a type II mAb binding to an exosite on APC (required for anticoagulant activity) as shown by X-ray crystallography. Both mAbs increase thrombin generation and promote plasma clotting. Type I blocks all APC activities, whereas type II preserves APC's cytoprotective function. In normal monkeys, type I causes many adverse effects including animal death. In contrast, type II is well-tolerated in normal monkeys and shows both acute and prophylactic dose-dependent efficacy in hemophilic monkeys. Our data show that the type II mAb can specifically inhibit APC's anticoagulant function without compromising its cytoprotective function and offers superior therapeutic opportunities for hemophilia.

[1] US Innovation Center, Bayer, 455 Mission Bay Blvd. South, San Francisco, CA 94158, USA. [2] Biological Research, Bayer AG, 42113 Wuppertal, Germany. [3] Beijing Advanced Innovation Center for Structural Biology, School of Life Sciences, Tsinghua University, Beijing 100084, China. [4] Structural Biology, Bayer AG, 13342 Berlin, Germany. [5] Biological Development, Bayer, Berkeley, CA 94710, USA. [6] Pathology/Toxicology, Bayer AG, 42096 Wuppertal, Germany. [7] The Scripps Research Institute, 10550 North Torrey Pines Rd., La Jolla, CA 92037, USA. [8] TRG-Cardiology/Hematology, Bayer AG, Aprather Weg 18a, 42113 Wuppertal, Germany. ✉email: xiao-yan.zhao@bayer.com; lmosnier@scripps.edu; volker.laux@ime.fraunhofer.de

Deficiency of coagulation factors (factor VIII [FVIII] or factor IX [FIX]) can result in recurrent bleeding in people with hemophilia (PWH)[1]. Treatment of PWH involves FVIII or FIX replacement products; however, 25–50% of persons with hemophilia A and 1–3% with hemophilia B develop inhibitors (i.e., neutralizing antibodies) against FVIII or FIX, respectively, which can render these products ineffective[2]. Blocking the endogenous anticoagulant pathways may offer novel therapeutic opportunities for PWH with inhibitors[3].

The protein C (PC) pathway critically regulates coagulation[4,5]. During coagulation, PC zymogen is cleaved to its active enzyme, APC, by thrombin in complex with thrombomodulin (TM) and endothelial cell PC receptor (EPCR) on the endothelial surface. APC functions as an anticoagulant by degrading activated factor V (FVa) and FVIIIa, thereby inhibiting thrombin formation. Bleeding disorders where hemostasis is deregulated may be treated by inhibition of APC to promote thrombin generation[6]. In fact, partial APC resistance reduces the frequency and severity of bleeding in PWH with the common FV Leiden variant, providing an early proof of concept in humans[7–9].

APC also exhibits cytoprotective effects, including anti-apoptotic and anti-inflammatory activities, as well as endothelial barrier stabilization[5]. These effects are mediated through EPCR and protease-activated receptor 1 (PAR1). APC protects animals from lipopolysaccharide or *Escherichia coli*-induced septic lethality. Furthermore, extracellular histones are major mediators of death in sepsis, and APC inhibits histone cytotoxicity via proteolytic cleavage[10].

Because the anticoagulant and cytoprotective activities appear to be located in distinct sites of the APC molecule[11–13], a protein therapeutic that selectively targets the anticoagulant activity while preserving the beneficial "cytoprotective" functions of APC would be desirable. Given the 1700-fold difference in plasma concentrations of APC (~40 pM) vs. PC (70 nM)[14], high specificity is required for antibodies to block APC function. Here, we identify two types of anti-APC mAbs: a type I active-site binder and a type II non-active-site binder. Both mAbs effectively increase thrombin generation in vitro by blocking the anticoagulant activity of APC, but only type II retains APC's cytoprotective function, and is safe to normalize hemostasis in monkeys with hemophilia. Thus, type II anti-APC mAb has therapeutic potential for prophylactic treatment of PWH with inhibitors.

## Results

**Binding specificities of type I and type II mAbs.** Type I anti-APC mAb was identified by panning the n-CoDeR® phage-display library of human antibody Fab fragments (BioInvent International AB). Candidate C25K23 was germlined and optimized to become TPP-2312 (BAY1316786). Type II was derived from HAPC1573[15] and humanized/optimized to become TPP-4885 (BAY1896502). Both type I mAb (TPP-2312 or BAY1316786) and type II mAb (TPP-4885 or BAY1896502) are hIgG2.

Surface plasmon resonance (SPR) analyses demonstrated specific binding of type I and type II to human APC (hAPC, $K_D$ 13 and 10 nM, respectively, Fig. 1a, b) and showed no binding to human PC (hPC) (Fig. 1c, d). Type I showed a slower on- and off-rate than type II. By ELISA (Fig. 1e, f), both mAbs showed dose-dependent specific binding to hAPC (half-maximal effective concentration [EC$_{50}$] at 0.4 and 0.8 nM, respectively) with no binding to hPC (type II) and some detectable binding to hPC at high concentrations (type I). High selectivity of these mAbs (i.e., no off-target binding) was also confirmed by ELISA because neither of them bind to other coagulation proteases, such as FIIa, FVIIa, FIXa, FXa, FXIa, and FXIIa (data not shown). Type I but not type II reduced hPC activation by thrombin–TM, consistent

with some residual affinity of type I for hPC (Supplementary Fig. 1).

Phe–Pro–Arg–chloromethylketone active-site-blocked APC (PPACK-hAPC) was used to investigate the binding epitopes of these mAbs relative to the active site of hAPC. Type II was able to bind to both PPACK-hAPC and hAPC, whereas type I no longer bound to PPACK-hAPC (Fig. 1g, h), suggesting that type I but not type II binds the active site of APC. As expected, the anti-APC-Gla-domain antibody R41C17 bound to both PPACK-hAPC and hAPC in this direct ELISA. Similar results were obtained when the type I and type II Fabs were used instead of IgG (Fig. 1h). These ELISA results were also confirmed by SPR (Supplementary Table 1).

**Type I and Type II mAbs bind to distinct epitopes on APC.** The crystal structure of hAPC–type I (C25K23) Fab complex was determined at 2.2-Å resolution (Fig. 2a; Table 1). The C25K23Fab mainly used its HCDR3 loop to interact with the APC catalytic domain (Fig. 2b; Supplementary Fig. 2a). The side chain of

**Table 1 Diffraction data collection and structural refinement statistics.**

| | (Type I) C25K23-APC | (Type II) h1573-APC |
|---|---|---|
| Data collection | | |
| Beamline | SSRF BL17U | SSRF BL17U |
| Wavelength | 1.000 Å | 1.000 Å |
| Space group | C2 | P4$_3$2$_1$2 |
| Cell dimensions | | |
| a, b, c (Å) | 112.257, 94.881, 114.298 | 124.339, 124.339, 666.156 |
| α, β, γ (°) | 90, 105.51, 90 | 90, 90, 90 |
| Resolution (Å) | 50.0−2.20 (2.26−2.20) | 50.0−3.70 (3.79−3.70) |
| $^aR_{merge}$ (%) | 9.6 (98.9) | 27.6 (0.00) |
| $^bR_{pim}$ (%) | 5.6 (53.1) | 14.9 (65.6) |
| $^cCC_{1/2}$ of the highest-resolution shell | 0.825 | 0.602 |
| I/σI | 13.7 (2.3) | 6.0 (1.5) |
| Completeness (%) | 96.9 (97.1) | 99.6 (99.8) |
| Redundancy | 3.1 | 5.5 |
| Refinement | | |
| Resolution (Å) | 28.7−2.20 | 33.4−3.7 |
| No. of reflections | 50,696 | 56,425 |
| $^dR_{work}/R_{free}$ (%) | 20.1/24.1 | 28.0/30.2 |
| No. of atoms | | |
| Protein | 5725 | 5636 |
| Water | 191 | |
| B-factors (Å$^2$) | | |
| Protein | 59.2 | 95.5 |
| Water | 56.6 | |
| r.m.s. deviations | | |
| Bond lengths (Å) | 0.008 | 0.004 |
| Bond angles (°) | 1.199 | 1.130 |
| Ramachandran plot (%) | | |
| Most favored | 86.7 | 80.3 |
| Additionally allowed | 12.1 | 17.3 |
| Generously allowed | 1.1 | 1.1 |
| Disallowed | 0.2 | 1.3 |

$^aR_{merge} = \sum_{hkl}\sum_j | I_j(hkl) - \langle I(hkl)\rangle|/\sum_{hkl}\sum_j I_j(hkl)$, where $I$ is the intensity of reflection and $h$, $k$, and $l$ are the indices of the reflections.
$^bR_{pim} = \sum_{hkl}[1/(N-1)]^{1/2}\sum_j | I_j(hkl) - \langle I(hkl)\rangle|/\sum_{hkl}\sum_j I_j(hkl)$, where $N$ is the redundancy of the dataset.
$^cCC_{1/2}$ is the correlation coefficient of the half-datasets.
$^dR_{work}$ and $R_{free}$ are defined by $R = \sum_{hkl}| |F_{obs}| - |F_{calc}| |/\sum_{hkl}|F_{obs}|$, where $F_{obs}$ and $F_{calc}$ is the observed and the calculated structure factor, respectively. $R_{free}$ is the cross-validation $R$ factor for the test set of reflections (5% of the total) omitted in model refinement.

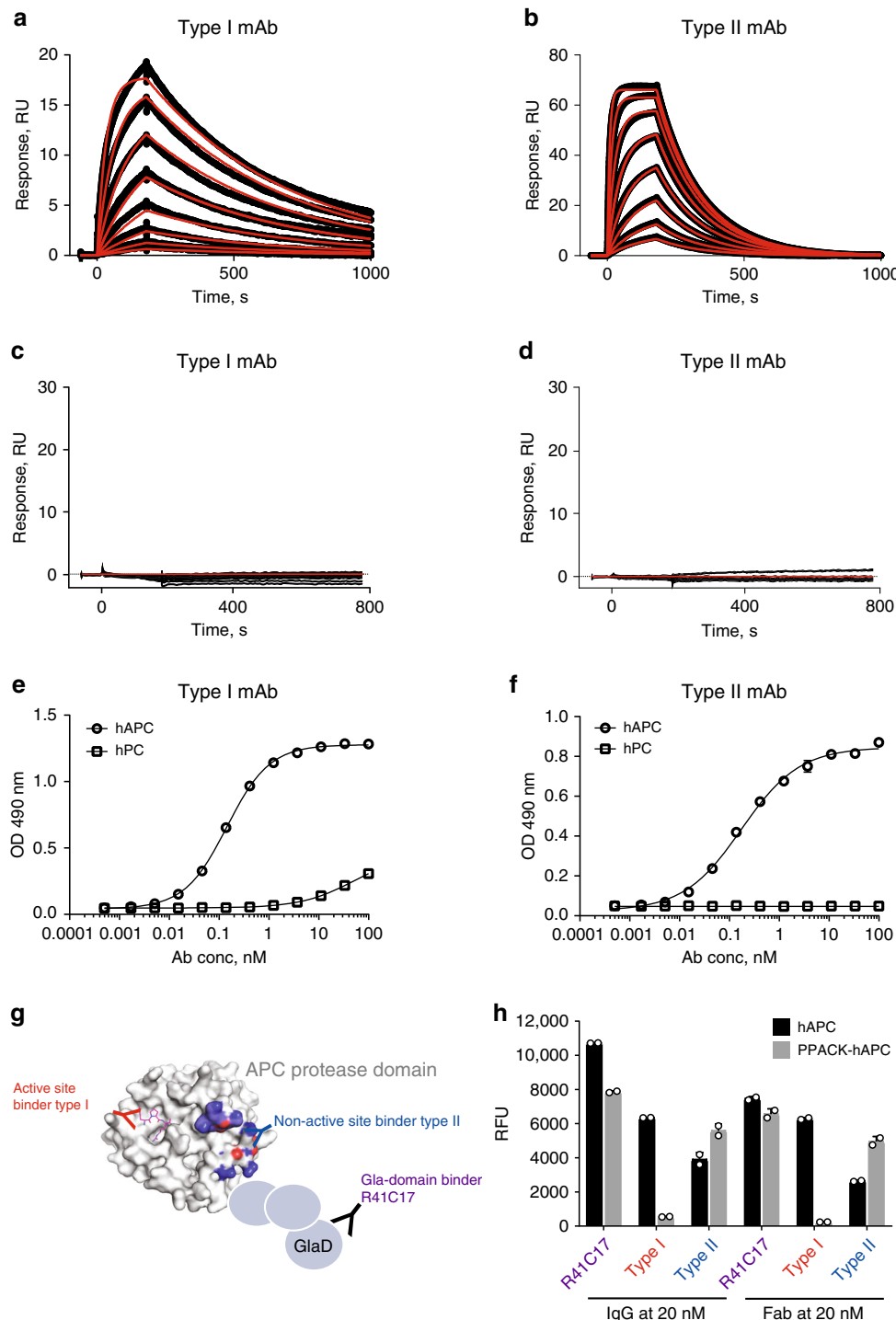

**Fig. 1 Antibody characterization using SPR and ELISA. a–d** Affinity measurement by SPR for type I and type II mAbs: Biacore sensograms showing the association and dissociation curves for antibody binding to hAPC at concentrations of 1.56, 3.125, 6.25, 12.5, 25, 50, 100, and 200 nM (**a**, **b**) or to hPC at concentrations of 1.56, 3.125, 6.25, 12.5, 25, 50, 100, 200, 400, and 1000 nM (**c**, **d**). Black traces represent experimental data, and red traces represent the corresponding fits. Type I showed a slower on-rate ($k_a$, $1.5 \times 10^5\,M^{-1}\,s^{-1}$) and off-rate ($k_d$, $1.9 \times 10^{-3}\,s^{-1}$) than type II ($k_a$, $6.1 \times 10^5\,M^{-1}\,s^{-1}$; $k_d$, $6.2 \times 10^{-3}\,s^{-1}$). **e**, **f** Antibody-binding specificity by ELISA for type I mAb (**e**) and type II mAb (**f**) bound to hAPC (circle) or hPC (square). ELISA results are shown as mean ± SDs from triplicate wells for each antibody concentration, and experiments were repeated >3 times. **g** Schematic representation of the binding sites on APC protease domain for type I, type II, and R41C17 mAbs. R41C17, an anti-APC Gla-domain antibody, was used as control. **h** Type II but not type I bound the complex of PPACK-hAPC. PPACK-hAPC or hAPC was coated onto a MaxiSorp plate at 100 ng per well overnight. IgG or Fab at concentrations starting at 20 nM were tested for binding. SPR surface plasmon resonance, ELISA enzyme-linked immunosorbent assay, hAPC human activated protein C, hPC human protein C, RU relative unit, OD490 absorbance at 490 nm, PPACK Phe–Pro–Arg–chloromethylketone, RFU relative fluorescence unit, IgG immunoglobulin G, Fab antigen-binding fragment.

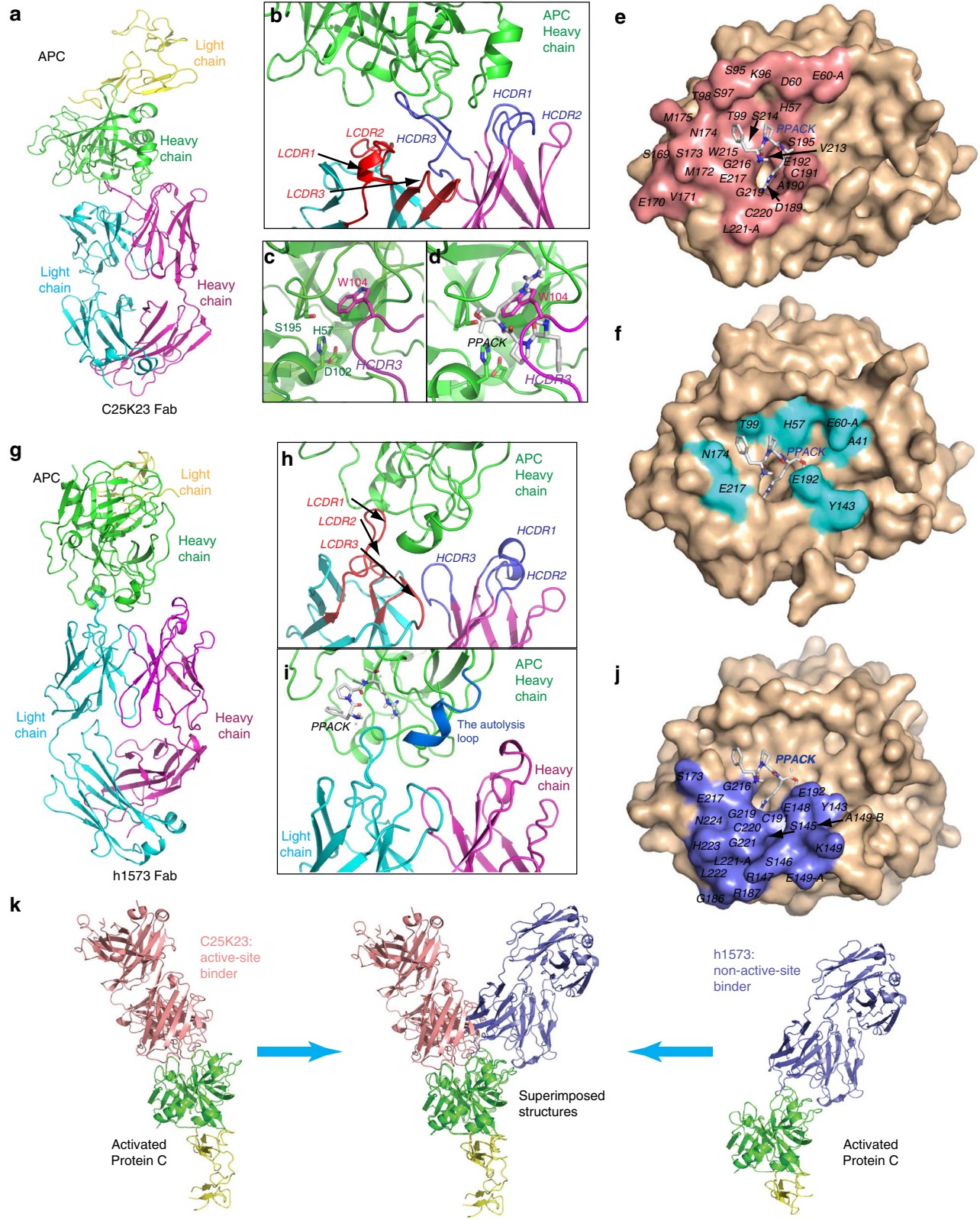

residue W104 of the HCDR3 inserted into the catalytic pocket of APC (Fig. 2c) and showed steric overlap with the tripeptide inhibitor PPACK in the previously reported[16] structure of active-site-inhibited APC (PDB code: 1AUT) (Fig. 2d). Thus, the binding epitope of C25K23Fab overlaps with the active site of APC (Fig. 2e, f, Supplementary Fig. 2c), suggesting that type I mAb is an active-site binder of APC.

The crystal structure of hAPC–type II (h1573) Fab complex was determined at 3.7-Å resolution (Fig. 2g; Table 1). The LCDR1 loop of the h1573Fab played a major role in binding to APC (Fig. 2h; Supplementary Fig. 2b). The binding of h1573Fab to APC did not have much steric overlap with PPACK (Fig. 2i). Upon h1573Fab binding, conformation of the His144–Thr152 loop (chymotrypsin numbering[16]) of APC (the autolysis loop)

**Fig. 2 Structural epitope mapping of type I and type II mAbs on APC by X-ray crystallography. a** Structure of the APC-C25K23Fab (type I) complex. **b** Close-up view showing that the HCDR3 loop plays a major role in interaction with APC. **c** Close-up view showing that the side chain of residue W104 of the HCDR3 inserts into the catalytic pocket of APC and **d** has steric overlap with the APC inhibitor PPACK. **e** Epitope of APC (salmon) within 5 Å to type I Fab marked on APC (wheat). **f** Critical residues (cyan) of APC interacting with PPACK (according to Fig. 5 of Mather et al.[16], PDB code: 1AUT) on APC (wheat). **g** Structure of the APC-h1573Fab (type II) complex. **h** Close-up view showing that the LCDR1 loop plays a major role in interaction with APC, and **i** the h1573Fab binding does not block the binding site of PPACK on APC, and the autolysis loop (His144–Thr152) of APC is observed in the APC-h1573Fab crystal structure (marine), but invisible in the APC–PPACK and APC-C25K23Fab structures (Supplementary Fig. 2d). **j** Epitope of APC (slate) within 5 Å to type II Fab marked on APC (wheat). **k** Side view of type I and type II Fab–APC complexes and structural superimposition of the two structures. In panels **d**, **e**, **i**, **j**, the positions of PPACK were determined by structural superimposition with APC–PPACK complex structure. APC residues are indicated using the Chymotrypsin numbering[16]. HCDR3 heavy-chain complementarity-determining region 3, LCDR1 light- chain complementarity-determining region 1, PDB protein data bank.

was fixed in contrast to the APC–PPACK complex and the hAPC–type I Fab complex (Fig. 2i, Supplementary Fig. 2d). The binding epitope of h1573Fab has little overlap with the active site of APC (Fig. 2f, j, Supplementary Fig. 2c). These results indicate that type II is a non-active-site binder of APC binding to the autolysis loop, and potentially residues L222–N224 (Fig. 2j). The autolysis loop comprises a known APC–FVa[17] interface, providing a likely explanation how type II inhibits APC's anticoagulant activity. Superimposition of the 2 APC–Fab complex structures (Fig. 2k) shows that unlike type I that sits inside the catalytic cleft of APC, type II contacts the autolysis loop and the mouth of the APC catalytic cleft.

In summary, type I and type II mAbs bind to APC using different epitopes and paratopes (Fig. 2e, j; Supplementary Fig. 2c, e). While the mAbs use different CDR loops to form major contacts with APC, both type I HCDR3 and type II LCDR1 are relatively long in length, consisting of 15 residues, providing a potential explanation for the selectivity of these mAbs for APC over PC.

**Both mAbs inhibit APC enzyme activity and are procoagulants.** Type I showed a complete (98%) inhibition of APC amidolytic activity with $IC_{50}$ of 4.8 nM (Fig. 3a; Supplementary Fig. 3), while type II showed only partial dose-dependent inhibition and reached a plateau at 10 nM with ~43% inhibition of APC activity. APC enzyme kinetics (data not shown) indicated that type I, but not type II, is a competitive inhibitor of APC.

In APC-mediated FVa inactivation assays, both mAbs dose-dependently inhibited proteolysis of FVa (Fig. 3b), indicating the ability of the mAbs to inhibit APC-mediated inactivation of its physiological substrate FVa.

Thrombin-generation assays (TGA) were used to demonstrate the in vitro activity of the anti-APC mAbs, and to determine the extent to which protecting FVa from APC-mediated proteolysis contributes to increased thrombin generation at the site of tissue damage. When TM was added to the plasma to promote APC generation, both mAbs increased thrombin generation in a dose-dependent manner with an $EC_{50}$ ~38 nM for both mAbs (Fig. 3c). The endogenous thrombin potential (ETP) and peak thrombin were increased from 50.6 to 469 nM min ETP and 6.17 to 48.4 nM peak by type I and from 54.5 to 223.6 nM min ETP and 8.85 to 27.4 nM peak by type II (Fig. 3c, d). Using endothelial cells as the physiologic surface for APC generation during coagulation, an inhibitory anti-TM antibody increased thrombin generation in the absence of exogenous tissue factor (TF) considerably (Fig. 3e), indicating that APC generation under these conditions was primarily dependent on TM. In this cellular TGA assay, both anti-APC mAbs also promoted thrombin generation up to twofold (Fig. 3f).

The effects of type I and type II mAbs on the clotting time of normal human plasma were determined by a modified aPTT assay, using Protac to rapidly activate PC. Protac-aPTT clotting

times in normal human plasma were reduced in a dose-dependent manner from 524 to 98 s (type I) and from 467 to 270 s (type II) (Fig. 3g), whereas isotype control IgG (M14–IgG2) had no effect. Inhibition of APC anticoagulant activity in normal plasma was greater by type I than by type II (93% vs. 50%, Fig. 3h); however, $IC_{50}$ was determined to be ~7 nM for both mAbs.

Similar results were obtained in hemophilic plasma (Fig. 3i, j, Supplementary Fig. 4) and in FVIII-antibody-treated normal human plasma (Fig. 3k, l). In hemophilia A plasma (Fig. 3i), clotting times were reduced from 437 to 175 s by type I and from 452 to 343 s by type II. Inhibition of APC activity (Fig. 3j) was greater by type I (max, 87%; $IC_{50}$, 5 nM) than type II (max, 42%; $IC_{50}$, 11.6 nM). Moreover, in FVIII-antibody-treated normal human plasma, $IC_{50}$ was 5 nM for type I and 12 nM for type II (Fig. 3k, l). The results of these different plasma assays demonstrated that both mAbs are APC-inhibition-mediated procoagulants, and that type I is more potent than type II.

**Distinct effects of mAbs on APC's cytoprotective function.** Based on the known effects of APC in reducing histone toxicity in vitro and in vivo[10], we tested the influence of both mAbs on APC activity using a histone-mediated toxicity assay with HUVEC cells (Fig. 4a). Type I blocked APC's protective activity and reduced live-cell numbers in a dose-dependent manner. In contrast, type II did not inhibit APC's protective activity and even demonstrated an increased protection at the highest concentration (300 nM) (Fig. 4a). These data indicate that type II did not inhibit APC's ability to reduce histone toxicity in vitro.

A SEAP–PAR1 reporter construct on transfected HEK293/wt-EPCR cells was employed to investigate the possible effects of the mAbs on APC-mediated cleavage of PAR1 that is required for many of APC's cytoprotective effects on endothelial cells[5]. Type I completely prevented APC-mediated PAR1 cleavage, whereas only a partial inhibition was observed by type II (Fig. 4b). Although type II at 100 nM resulted in approximately 40% of normal APC-mediated SEAP–PAR1 cleavage on HEK293/wt-EPCR, type II did not block APC-mediated PAR1 cleavage on EA. hy926 endothelial cells (Fig. 4c). In contrast, type I inhibited cellular PAR1 cleavage by APC completely.

To understand the consequences of the anti-APC mAbs on APC-mediated PAR1 cleavage for APC's cytoprotective activities, APC-mediated endothelial barrier stabilization was determined in real time by measurement of changes in transendothelial electric resistance (TER) upon thrombin-induced barrier disruption. Under conditions where the mAbs inhibited >75% of APC's anticoagulant activity (Supplementary Fig. 5), type II did not affect APC's ability to protect against barrier disruption by thrombin (Fig. 4d). In contrast, type I abolished barrier protection by APC. Furthermore, type II at 1500 nM (30-fold excess over APC) partially reduced endothelial barrier function of APC, reaching 55% of normal barrier protection (Fig. 4e),

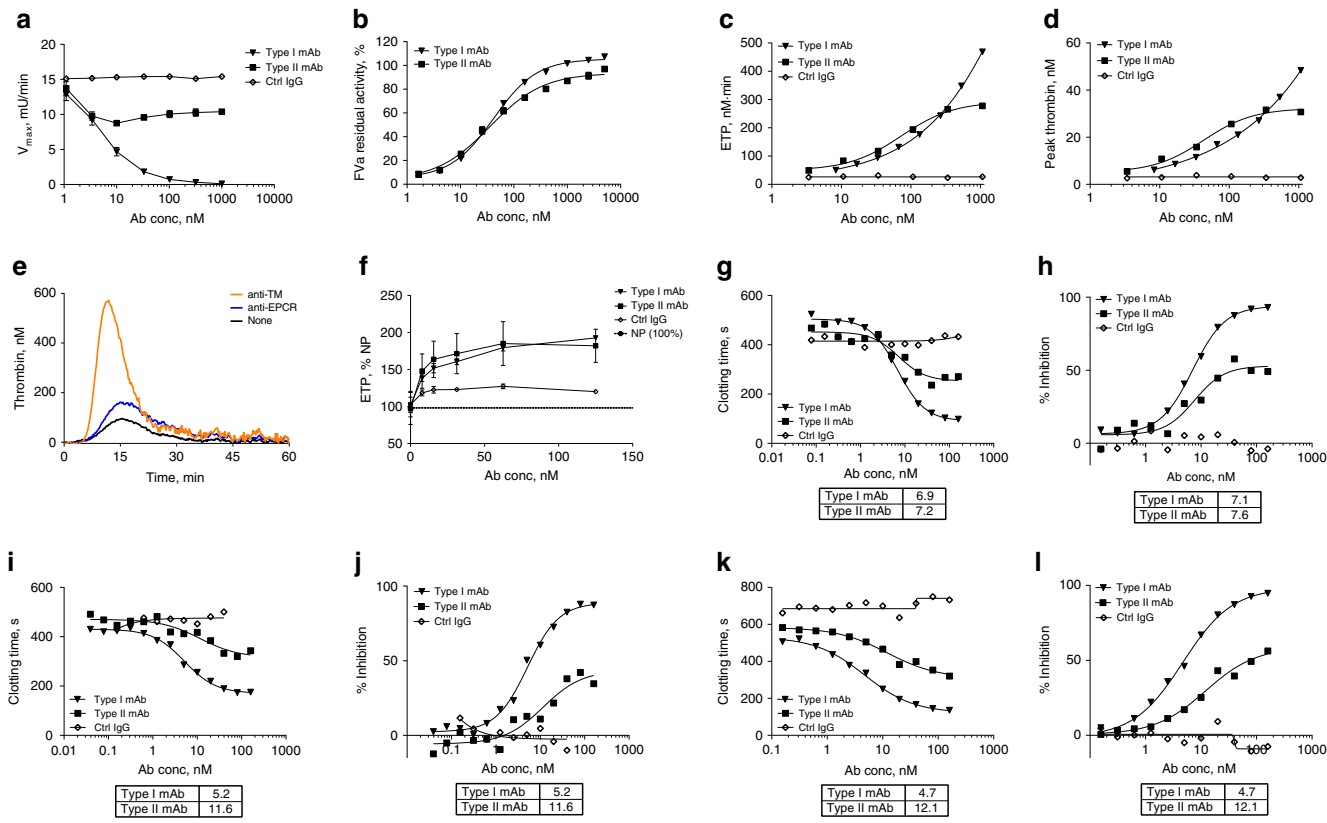

**Fig. 3 Type I and type II mAbs are procoagulants in vitro. a** APC amidolytic activity is defined by the maximum velocity ($V_{max}$) of its hydrolyzing reaction toward the small chromogenic substrate spectrozyme PCa, and percent inhibition of APC amidolytic activity by mAbs is shown in Supplementary Fig. 3. Data are expressed as mean ± SD. $N$ = 4–6. **b** Both mAbs protect FVa from APC-mediated proteolysis. In the absence of APC, the activity of FVa in the prothrombinase assay was designated as 100%. In the absence of added antibodies, APC proteolyzed FVa and reduced its activity to 6.4 ± 0.3% (mean ± SD). Assays were repeatedly performed in triplicates; a typical curve is shown. **c**, **d** TM–TGA in normal human plasma (FACT): **c** ETP and **d** peak thrombin levels. **e**, **f** TGA on EA.hy926 endothelial cells: **e** baseline thrombin levels (nM) shown as a representative experiment and **f** ETP as %NP (percent of normal pooled plasma). Data represent mean values ± SD from three independent experiments. **g–l** Plasma-clotting assays where IC50 values in nM of mAbs are compared in a table below each panel. Protac-aPTT using FACT: **g** clotting time in seconds (s), **h** percent inhibition. Protac-aPTT using HemA plasma: **i** clotting time in seconds (s) and **j** percent inhibition. Protac-aPTT using FVIII Ab-treated FACT: **k** clotting time in seconds (s) and **l** percent inhibition. In **c**–**l**, experiments were run with replicate samples in each assay, and data reflect the average of two replicates. Assays were repeated ≥3 times. FACT normal human plasma, Protac a protein C activator from *A. contortrix* venom.

consistent with a similar partial inhibition of APC's amidolytic activity by type II.

Since exosites, including the autolysis loop on APC, are often shared in part among different substrates[18–20], the effect of type II on the half-life of APC in human plasma was determined. Type II prolonged the $t_{1/2}$ of residual APC amidolytic activity in plasma from 20 to 58 min (Fig. 4f), indicating that type II also delays the inactivation of APC by serine protease inhibitors (SERPINs) in plasma, which is consistent with previous implication of the autolysis loop (part of type II-binding epitope) in reactions with SERPINs[19].

**Acute and prophylactic efficacy of type II mAb in monkeys**. In order to identify animal models for in vivo testing, species cross-reactivity of these mAbs was examined by ELISA. Type I bound to APCs of monkey, rabbit, and dog (with EC50 at ~1 nM), and showed some residual binding to monkey PC similar to hPC. In contrast, type II only cross-reacted with monkey APC (with EC50 at ~1 nM) and showed no detectable binding to monkey PC (data not shown).

In TM–TGA assays, both type I and type II dose-dependently increased ETP (Fig. 5a) and peak thrombin concentration (Fig. 5b) in pooled cynomolgus monkey plasma. In FVIII-

antibody-induced hemophilia A (HemA) monkey plasma, both mAbs reduced Protac-aPTT clotting time in a dose-dependent manner (Fig. 5c). Both mAbs inhibited APC anticoagulant activity (Fig. 5d), but type I (IC50, 9.3 nM) was more potent than type II (IC50, 19.3 nM). In normal monkeys, type I induced 50% plasma PC depletion 24 h post dosing in all dose groups (10, 30, and 100 mg/kg) (Fig. 5e), except 3 mg/kg (Supplementary Fig. 6a), and caused significant reduction in plasma FVIII levels in high-dose monkeys (30 and 100 mg/kg) (Supplementary Fig. 7a). One monkey died 45 min post dosing of type I (30 mg/kg). Other adverse events included activation of the endothelium in liver and lung, neutrophil beading, leaky vessels, and hematomas (data not shown), and required the unscheduled euthanasia of the three remaining monkeys per protocol. In contrast, type II was generally well-tolerated and did not cause a depletion of PC or FVIII (Fig. 5f; Supplementary Fig. 6b and 7b). Furthermore, type II has a considerably longer half-life (~289 h) than type I (~111 h) when dosed at 3 mg/kg iv (Fig. 5g; Supplementary Table 2).

To evaluate the acute therapeutic effect of type II, a severe acquired HemA monkey bleeding model was developed using a sheep anti-human FVIII polyclonal antibody to induce HemA followed by anti-APC mAb treatment and tail-vein injury. Administration of the anti-FVIII antibody (2.8 mg/kg) reduced plasma FVIIIa activity from a baseline of ~1.5 IU/mL to below the

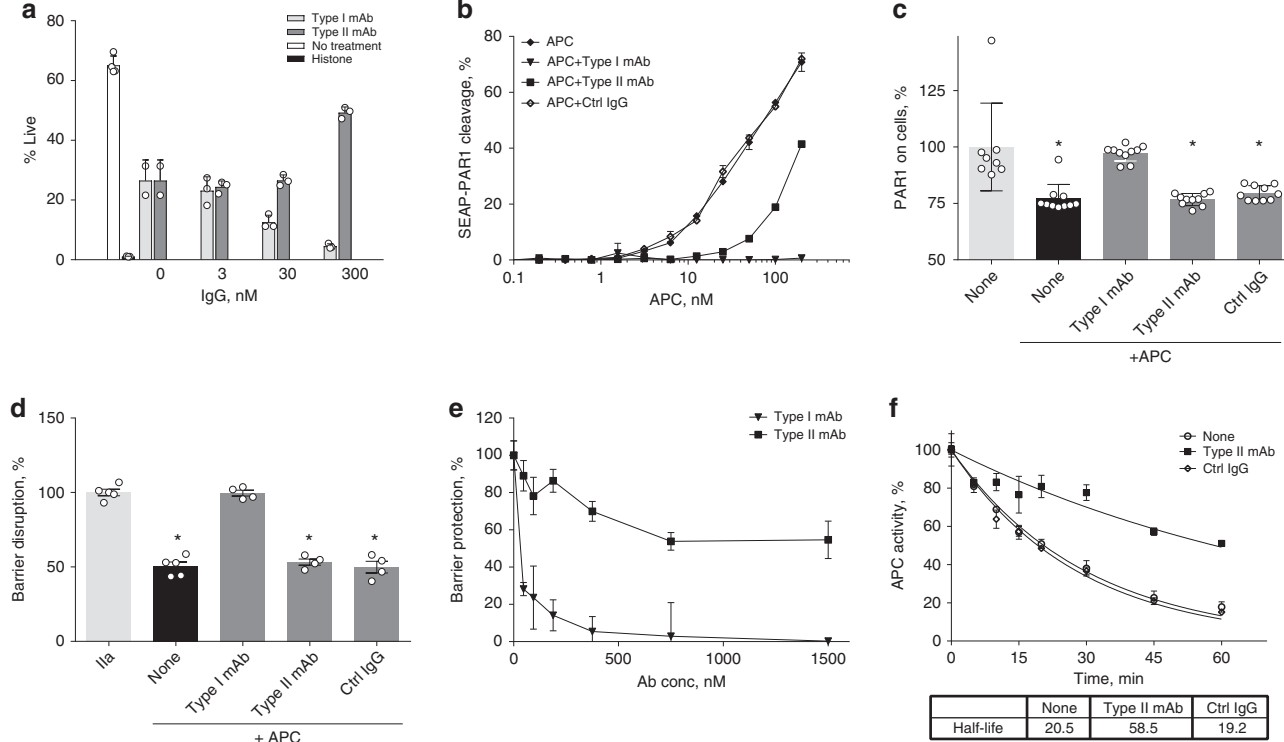

**Fig. 4 Distinct effects of type I and type II mAbs on the cytoprotective activities of APC. a** Histone-mediated cytotoxicity assay using HUVECs (% live cells after 2 h). Positive (50 μg/mL histone 3, 0% live cells, black bar) and negative (no histone 3, 70% live cells, open bar) controls. Reduction of histone cytotoxicity by hAPC (20 nM, gray bars) in the presence of type I mAb (light gray) or type II mAb (dark gray). The results are shown as mean + SD of three independent experiments. **b** Effect of mAbs on PAR1 cleavage of SEAP–PAR1 reporter construct on transfected HEK293/wt-EPCR cells. Shown are mean ± SEM of $n = 4$ performed on two independent cell seedings. **c** PAR1 cleavage on EA.hy926 endothelial cells. The ATAP antibody reports PAR1 cleavage at Arg46[50]. Shown are mean ± SD of $N = 10$. * denotes $p < 0.05$ tested with ANOVA with Dunnett's multiple-comparison test in both (**c**) and (**d**). **d**, **e** APC (50 nM)-mediated endothelial barrier protection in response to thrombin (2 nM)-induced barrier disruption of an EA.hy926 endothelial cell monolayer was determined using the iCelligence system. **d** Permeability is expressed as the percentage of maximal barrier disruption induced by thrombin (100%) in the absence of APC. Shown are mean ± SEM of $N = 4$. **e** Effect of the mAbs on APC's barrier protection is expressed as the percentage of the maximal barrier-protective effect of APC in the absence of mAbs. Shown are mean ± SEM of $N = 3$. **f** Effect of the type II mAb on the inactivation of APC in human plasma by SERPINs. Data represent mean values ± SD from at least three independent experiments. HUVEC human umbilical vein endothelial cells, SEAP–PAR1 secreted embryonic alkaline phosphatase (SEAP) fused to the N terminus of human PAR1, HEK293/wt-EPCR HEK293 cells with stable expression of wild-type human EPCR, SERPINs serine protease inhibitors.

lower limit of quantification (LLOQ at 0.04 IU/ml, Fig. 5h), and led to a significantly longer bleeding time (from 400 to 1100 s) (Fig. 5i). Recombinant FVIIa (270 μg/kg), a clinically relevant positive control, partially corrected the prolonged bleeding time. Type II significantly and dose-dependently reduced bleeding times, with the 10 mg/kg type II mAb restoring the bleeding time back to normal (~400 s, Fig. 5i).

To evaluate the prophylactic efficacy of type II, we used anti-FVIII mAb (BO2C11[21,22]) to induce long-term hemophilia in cynomolgus monkeys. Administration of >1 mg/kg of BO2C11-cynoIgG1 significantly prolonged aPTT in these monkeys for at least 72 h (Supplementary Fig. 8a) and led to hemorrhagic anemia along with a significant reduction in hemoglobin (Supplementary Fig. 8b), which was worsened by an intramuscular needle injury (Supplementary Fig. 9). Prophylactic administration of type II mAb 7 days before induction of HemA ($t = 0$) by 0.1 mg/kg BO2C11-cynoIgG1 and intramuscular needle injury ($t = 2$ h) suggested an apparent dose-dependent bleed protection as evident by the correlation between the type II dose and the reduction in anti-FVIII-induced hemorrhagic anemia ($N = 9$, Fig. 5j). Two monkeys in the saline-treated group ($N = 6$) died because of severe blood loss before 72-h sampling, whereas none of the animals in the type II-treated groups died, further

supporting a relevant prohemostatic effect in type II-treated animals.

## Discussion

Blocking the natural anticoagulants (i.e., endogenous inhibitors of coagulation) is emerging as a promising strategy for hemophilia therapy, which is supported by initial results of clinical trials[23,24]. The PC pathway is one of the physiological important natural anticoagulant pathways, and initial studies in mice suggest that targeting this pathway promotes hemostasis in hemophilia[6,25]. But targeting the PC pathway does come with its specific challenges. APC is a multifunctional protease, mediating both anticoagulant activity and broad cytoprotective activities on multiple cells, and these cytoprotective activities are essential to maintain the integrity of the vasculature, including that of the blood–brain barrier[26,27]. Furthermore, purpura fulminans due to severe PC deficiency indicates the potential risks associated with lowering PC levels[28,29]. Thus, a prohemostatic approach targeting the PC pathway in hemophilia should be specific for APC without affecting circulating PC levels, and should be selective for APC's anticoagulant activity without abrogating APC's cytoprotective activities. Previously, an engineered α1-antitrypsin variant with improved reactivity with APC was shown to reduce bleeding in

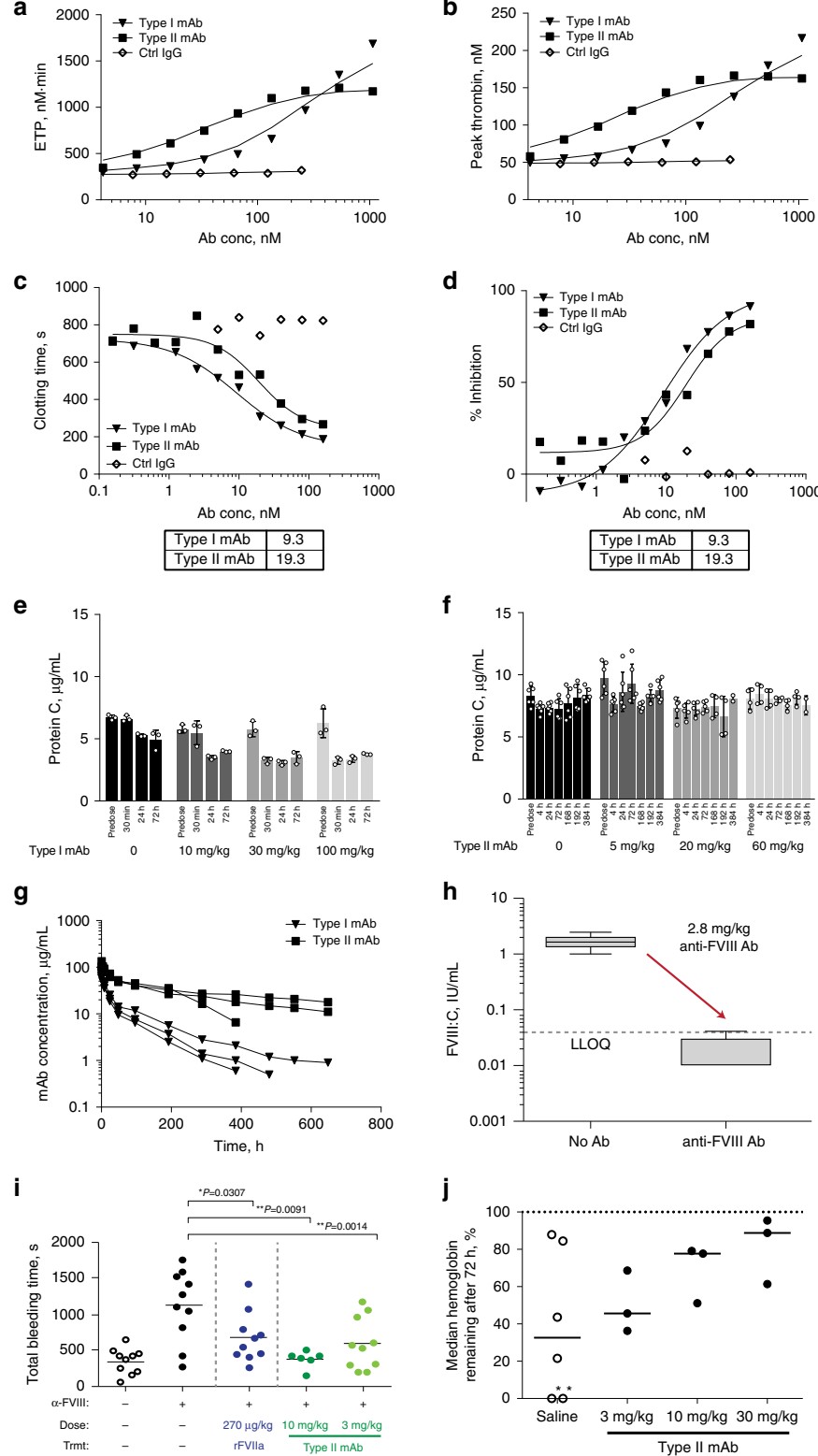

hemophilia B mice[6]. While this α₁-antitrypsin variant in principle inhibits all APC activities, selectivity for APC's anticoagulant activity was argued but not shown, based on different kinetics of APC inhibition and dissociation of the APC–EPCR complex[30]. Here, we identified an anti-APC type II inhibitory mAb that is uniquely specific for APC and highly selective for APC's anticoagulant activity over its cytoprotective functions, and showed that this type II mAb can be beneficial for hemophilia therapy.

Administration of the type II mAb in efficacy studies showed dose-dependent acute and prophylactic efficacy in curbing bleeding in hemophilic monkeys. The type II mAb at both 3 and 10 mg/kg was efficacious in curbing bleeding in monkey HemA model, and the latter dose restored the bleeding time back to normal. In exploratory safety studies in normal monkeys, no effects on circulating PC levels, and no mortality, clinical signs, or effects on clinical pathology (including coagulation parameters

**Fig. 5 Characterization of type I and type II mAbs in monkeys. a, b** TM–TGA in monkey plasma: **a** ETP and **b** peak thrombin levels. **c, d** Protac-aPTT in FVIII Ab-treated monkey plasma: **c** clotting time with $IC_{50}$ values of 9.3 vs. 19.3 nM and **d** % inhibition with $IC_{50}$ values. Experiments were performed with triplicate samples in (**a, b**) and duplicate samples in (**c, d**); data points reflect their average in each assay, and assays were repeated ≥3 times. **e–j** Testing mAbs in vivo. Plasma PC levels (µg/mL) in type I-treated (**e**) and in type II-treated (**f**) wild-type cynomolgus monkeys. Each bar represents the mean value ± SD of 3–6 measurements from one monkey (**e**) and two monkeys (**f**). **g** Pharmacokinetic profiles (c/t curves) of both mAbs at 3 mg/kg administered intravenously in wild-type cynomolgus monkeys. **h, i** Development of severe HemA monkey bleeding model and testing acute efficacy of type II mAb. **h** Plasma FVIII activity (IU/mL) in monkeys (n = 10 independent animals) before and after treatment with anti-FVIII antibody (no Ab vs. anti-FVIII Ab). LLOQ means the lower limit of quantification (0.04 IU/mL). Box and whiskers are based on the Tukey's method with the center line representing median, box representing interquartile range (IQR) between 25th and 75th percentiles, and whiskers representing 25th or 75th percentile plus 1.51 IQR. **i** Total bleeding time in second(s) post injury in normal monkeys (i.e., no treatment: Trmt−) and in HemA monkeys (i.e., treated with anti-FVIII antibody: α-FVIII), as well as in HemA monkeys treated with type II mAb at 3 and 10 mg/kg or with the known procoagulant rFVIIa (recombinant activated factor VII) at its pharmacological dose (270 µg/kg) as control. p Values < 0.05 are indicated (one-way ANOVA with Dunnett's multiple-comparison test). N = 10 monkeys per group. **j** Median hemoglobin remaining, 72 h post administration of BO2C11-cynoIgG1 (t = 0) and intramuscular needle injury (t = 2 h), as a readout of prophylactic efficacy of type II mAb in HemA cynomolgus monkeys. Median values of hemoglobin remaining in each group as % of non-hemophilic animals are shown by lines. * in the saline group denotes two monkeys that died.

[PT, aPTT, thrombin, and fibrinogen]) were observed with the type II anti-APC mAb. Notwithstanding the demonstrated safety of type II in exploratory monkey studies, future studies are needed to determine potential impact of the type II antibody on APC cytoprotective effects in vivo during inflammatory and/or ischemic challenge. The importance of maintaining APC's "cytoprotective" functions was highlighted by the adverse effects and animal death observed with administration in monkeys of the type I mAb that occupies the active site of APC, thereby inhibiting both anticoagulant and cytoprotective functions. Furthermore, as predicted by the detectable binding of hPC at high concentrations by ELISA, the type I mAb demonstrated a marked reduction in PC levels in monkeys, supporting the need for APC specificity.

X-ray crystallography revealed that type I and II mAbs bind to different epitopes on APC, leading to different functional consequences in vitro and in vivo. The potent anticoagulant activity of APC involves rapid cleavage at Arg506 in FVa[18,31]. This cleavage requires a positively charged surface on the protease domain of APC for normal interactions with FVa that includes loop 37, the $Ca^{2+}$-binding loop, and the autolysis loop[16,17,19,32–34]. The type II mAb interacted with a FVa-binding exosite of APC that included the autolysis loop[17], and its inhibition of APC's anticoagulant activity is likely due to interference with the APC–FVa interaction. Thus, the type II mAb induced endogenous APC to be cytoprotective-selective, similar to the 3K3A–APC variant that is known to retain normal cytoprotective function with reduced (>90%) anticoagulant function of APC[35].

In conclusion, we show that specific inhibition of the APC anticoagulant function by an inhibitory mAb is likely a viable, efficacious, and safe option for a bypass agent in hemophilia therapy.

## Methods

**Materials.** Recombinant human APC (hAPC, drotrecogin alfa [activated]) was purchased from Eli Lilly (Indianapolis, IN, USA); plasma-derived hAPC and hPC from Enzyme Research Laboratories (South Bend, IN, USA); calf thymus histones from Sigma-Aldrich (St. Louis, MO, USA); TM and human coagulation FVa from Hematologic Technologies (Essex Junction, VT, USA); Protac, a rapid activator of PC from Sekisui Diagnostics (Lexington, MA, USA); hAPC substrate Spectrozyme PCa from American Diagnostica (Pfungstadt, Germany) and Sekisui Diagnostics (Lexington, MA, USA), and S-2366 from Diapharma (West Chester, OH, USA).

**Panning for anti-APC type I mAb.** Panning and screening of a fully human fragment antigen-binding (Fab) antibody library (BioInvent International AB, Sweden) against hAPC were performed using the methods below[36].

(1) Library selections with biotinylated APC (in-solution panning): Two negative selection methods were carried out in parallel for the depletion of either nonspecific binders or nonspecific and PC binders. Dynabeads M280 Streptavidin was incubated with 100 nM biotin–TF (for nonspecific depletion) or 100 nM

biotin–PC (for nonspecific and PC depletion) and captured by a magnetic device. About $1–7.5 \times 10^{12}$ CFU Fab library phage, preblocked with DPBS/3% bovine serum albumin (BSA)/0.05% TWEEN, was incubated with biotin–TF or biotin–PC-coupled streptavidin beads on a rotator at room temperature (RT) for 2 h. The biotin–TF or biotin–PC streptavidin beads were captured and discarded. The resulting phage supernatants were incubated with 100 nM (first round), 50 nM (second round), or 10 nM (third round) biotin–APC in 1 ml of DPBS/3% BSA/0.05% TWEEN 20/1 mM $CaCl_2$ for 2 h at RT or 40 °C overnight. About 100 µl of streptavidin-coupled magnetic beads were added to the phage–APC solution and incubated for 30 min at RT. The phage–APC complex beads were captured on a magnetic device and washed with DPBS with 3% BSA or 0.05% TWEEN 20, depending upon the panning round. The bound phage was eluted with 1 mg/ml trypsin and neutralized with aprotinin. The eluted phage was then used to infect 10 ml of exponentially growing E. coli HB101F′ and amplified for the next round of selection. The phage stock was also analyzed by CFU titration (panning output).

(2) Library selections with immobilized APC (solid-phase panning): Similar to the in-solution panning, the phage library was precleared with biotin–TF or biotin–PC streptavidin beads to eliminate nonspecific binders or nonspecific and PC binders, respectively. The resulting phage supernatant was added to 96-well plates coated with 400 ng/well recombinant APC in DPBS at 4 °C overnight and incubated on a shaker for 1–2 h at RT. Unbound phage was washed away with DPBS with 3% BSA or 0.05% TWEEN 20, depending upon the panning round. The bound phage was eluted with 1 mg/ml trypsin and neutralized with aprotinin. The eluted phage was then used to infect 10 ml of exponentially growing E. coli HB101F′ and amplified for the next round of selection. The phage stock was also analyzed in a CFU titration (panning output).

DNA sequencing was performed on the positive antibody clones, resulting in 10 unique antibody sequences. Identical heavy-chain complementarity-determining region 3 (CDR3) sequences are found in 5 Fabs (C7I7, C7A23, T46J23, C22J13, and C25K23). The purified Fabs were characterized by a panel of functional assays to assess (1) their binding specificity (APC vs. PC), binding affinity (by ELISA and SPR), and species cross-reactivity (i.e., binding to APCs of different species' origins, including human, monkey, dog, rabbit, and mouse); (2) their binding selectivity against other vitamin K-dependent coagulation factors (e.g., FIIa, FVIIa, FIXa, FXa, and FXIIa); (3) their potency of inhibiting APC's anticoagulant activity in the plasma-clotting assay activated thromboplastin time (aPTT); (4) their effect on APC's protease enzymatic activity in buffer using amidolytic activity assay (on a small peptide substrate) and FVa inactivation assay (on the protein substrate FVa).

**Humanization of mouse anti-hAPC antibody HAPC1573 (mIgG1/k).** In order to generate anti-APC type II mAb, we started to humanize and optimize a mouse anti-hAPC antibody (HACP1573). HAPC1573 hybridoma cell line was kindly provided by Charles T. Esmon, PhD, at the Oklahoma Medical Research Foundation (Oklahoma City, OK, USA)[15]. Cells were maintained in RPMI1640 medium supplemented with 10% fetal bovine serum and subcloned in 96-well plates using the half-dilution method to gain clonality. Top titer producers were identified by ELISA and POROS-G (Thermo Fisher Scientific, Waltham, MA). Total RNA was extracted from $5 \times 10^6$ cells with mirVana miRNA isolation kit (cat#AM1560) from Ambion (Thermo Fisher Scientific) following the standard protocol. The purified RNA concentration was 0.6 µg/µL with 260/280 ratio of 2.0 and was stored at −80 °C. SuperScript 1-step reverse transcription polymerase chain reaction (RT-PCR) with Platinum Taq (Invitrogen, Carlsbad, CA, USA, cat#10928-034) was used for RT-PCR. The Ig-Primer set was from Novagen (EMD Chemicals, San Diego, CA, USA, cat#69831-3). The six primers for mouse immunoglobulin G (mIgG) V region heavy chain (VH) paired with MulgVH3′-2 and 7 for V region light chain (VL) paired with MulgVL3′-1 were used to identify the transcripts of HAPC1573. cDNA synthesized at 48 °C for 30 min followed by 30 cycles of amplification (95 °C × 15 s, 50 °C × 1 min, and 72 °C × 2 min) and 6-min extension at 72 °C. The RT-PCR

products at the correct size were sent for DNA sequencing analysis. For VH, only one RT-PCR product was generated when using the primer pair of MulgVH5′-B and MulgVH3′-2. DNA sequencing results matched completely with the predicted DNA sequences derived from the amino acid sequence analysis. For VL amplification, three pairs of primers generated products at the correct size: MuIgkVL5′-B, C, and G paired with MulgVL3′-1. The PCR products were named HAPC1573VL-B, C, and G, respectively. DNA sequencing analysis revealed that HAPC1573VL-B and C were identical, and they matched the expected sequence based on amino acid sequence analysis.

HAPC1573 antibody humanization and optimization were as follows[37–40]. The humanization design was done using the following method: the CDR residues within the light chain and the heavy chain (VL/VH CDRs), respectively, were determined and annotated with Kabat numbering system (worldwide web at bioinf.org.uk/abs/#kabatnum). The canonical structures of the VH/VL CDRs were determined based on reports in the literature[39,40]. The sequence of HAPC1573-IgG was used to perform a PDB database blast search in order to obtain known antibody structures sharing the highest sequence identities with the antibody of interest. Based on this blast search, antibody sequences 1M71, 1M7D, and 1M7I were selected as template for modeling the heavy chain of TPP-1573, while 1IQW, 1IT9, and 2GCY were selected as template for light-chain modeling.

For building sequence homology models for the two chains, the Schrodinger suite software was used. Sequence models generated with this software tool were further analyzed in order to identify all residues in framework regions that potentially interact with residues from CDR regions.

Based on the output of software and visual inspection with the model, the following residues in framework were identified as residues that contribute to the supporting of CDR loops. Those were the light-chain residues Asp70, Tyr36, Thr69, Phe71, Ile2, and Tyr49, and for the heavy- chain residues Arg94, Arg38, Glu46, Trp47, Asp73, Arg71, and Trpl02. For the design of humanized VH, residues supporting loop structures and VH/VL interface were identified (International Application No. WO2008021 156).

Those amino acids that could have an impact on loop conformation and VH/VL interface were back-mutated. Then the VH sequences with the back mutations were aligned with the selected germline subfamily. The identities and similarities to each individual human germline framework sequences within the same canonical subsets were analyzed, and the germline sequence with the best overall homology to the murine VH sequence was identified. This sequence was selected as the acceptor human germline framework for grafting VH CDRs. Additional considerations for mutations included a Q1E mutation used to eliminate N-terminal pyroglutamate formation. Mutations also included those to maintain consensus within the selected VH family, for CDR canonical structures and VH/VL interface. Mutations also include those identified as within from the CDR-binding region according to molecular modeling. Analysis was performed to make sure that no N-linked glycosylation pattern (N–{P}–S/T) was found in the proposed humanized construct. The human JH region was selected based on best sequence homology. The same procedure as for the heavy chain was used to humanize the light chain.

### Optimization of anti-APC type I mAb.

For the optimization of C25K23, a similar approach was used in order to define the CDRs within the heavy and the light chain. Furthermore, a PDB database blast search was performed in order to obtain known antibody structures sharing the highest sequence identities with the antibody of interest. By exchanging those amino acids that differ from this reference sequence, framework sequences were adapted. In order to improve the binding activity of C25K23 to APC, all amino acids within the CDRs were exchanged by the other 19 possible amino acids by side- directed mutagenesis. These variants were then tested for binding activity. Those exchanges, which led to an improved binding activity, were combined. The outcome of the optimization approach is TPP-2312, which compared with C25K23, exhibits 2 distinct exchanges: one at position 56 within the CDR2 of the heavy chain (S56A), a second one in the CDR2 of the light chain (K55R).

### Expression and purification of type I Fab (C25K23Fab).

Recombinant anti-APC human Fabs (C25K23Fab, TTP-2781, Supplementary Table 3) were expressed in *E. coli* and purified to homogeneity by Protein A chromatography. Purified Fabs were shown to have a >90% purity and no noticeable aggregation determined by sodium dodecyl sulfate polyacrylamide gel electrophoresis (SDS-PAGE) and by analytical size- exclusion chromatography. EC$_{50}$ values of C25K23Fab binding to human APC and Gladomainless APC were 2–4 nM as measured by an APC-binding ELISA.

### Expression and purification of type II Fab (hu1573Fab).

The coding DNA for hu1573Fab (TPP-5040) sequences (Supplementary Table 3) was synthesized at GeneArt (Thermo Fisher Scientific), cloned into pTT5 (National Research Council Canada), and sequence-verified. The expression construct was transiently transfected into QMCF cells (a Chinese hamster ovary [CHO]-based cell line from Icosagen, San Francisco, CA, USA) with polyethylenimine. Qmix1 serum-free medium was used for expression. After transfection, cells were incubated at 37 °C for 2 days, then switched to 30 °C for another 4 days. The conditioned medium was

harvested 6 days post transfection by centrifugation and filtration. The expression level of hu1573Fab was checked by SDS-PAGE. The estimated yield by comparing with positive controls in Coomassie blue staining gel was about 80 mg/L.

One liter of serum-free cell-conditioned media was loaded onto a 5-mL anti-IgG light-chain affinity column (HiTrap KappaSelect, GE HealthCare Life Sciences, Little Chalfont, UK), pre-equilibrated in phosphate-buffered saline (PBS), at a flow rate of 0.8 mL/min using an Äkta Explorer system (GE HealthCare Life Sciences). UV absorbance was monitored at 280 nm. Unbound material was washed from the column with PBS, until a stable baseline was achieved. Fab protein was eluted using a short linear gradient 0–100% "Buffer B" over 10 min at a flow rate of 1 mL/min while collecting 1-mL fractions. "Buffer A" was PBS and "Buffer B" was 100 mM glycine, pH 2.5. Peak fractions were pooled, concentrated to 2 mL, and further purified by size-exclusion chromatography [SEC] using a SD-200 26/60 column (GE HealthCare Life Sciences). The mobile phase was PBS and the flow rate was 1 mL/min. Fractions (1 mL) were collected beginning 30 min after column loading. Peak fractions containing purified Fab were pooled, and the final protein was analyzed by SDS-PAGE, analytical SEC, and liquid chromatography mass spectrometry (LCMS). Binding activity of APC was measured by ELISA and SPR. The purity was quantified by analytical SEC (>98% purity monomer).

### Expression and purification of anti-hAPC mAbs (hIgG2).

When converting Fabs to full-length antibodies, hIgG2 isotype was used to minimize antibody effector functions. The sequences of full-length antibodies were first generated in silico by attaching the variable region sequences from Fab to the entire constant region sequences (hIgG2 CH1–CH2–CH3). Genes encoding the antibody heavy and light chains (see amino acid sequences of heavy and light chains in Supplementary Table 3) were subcloned separately in expression vector pTT5 (National Research Council, Canada) and cotransfected into HEK293-6E cells[41] (National Research Council, Canada) that were maintained in F17 medium (Invitrogen) supplemented with 4 mM GlutaMAX (Invitrogen), 0.1% Pluronic F-68 (Sigma-Aldrich), and 25 µg/mL G418 (Invitrogen). After 5–8 days of transient expression, the cleared supernatant was used to affinity-purify the antibody on an Äkta Purifier System (GE Healthcare) using a 10-mL HiTrap MabSelect Sure protein A column (GE Healthcare). Antibodies were eluted in 2 steps with 50 mM sodium acetate and 500 mM NaCl at pH 3.5 and pH 3.0. Combined elution fractions were neutralized using appropriate volumes of 2.5 M Tris base (pH 11). Aggregation products were removed by preparative size-exclusion chromatography on an Äkta Purifier System (GE Healthcare) using a custom-made Superdex 200 50/600 column (GE Healthcare), with a mobile phase of PBS (pH 7.4) at a flow rate of 6.0 mL/min.

### Production of anti-FVIII BO2C11-cynoIgG1 chimeric antibody.

BO2C11-cynoIgG1 (TTP-5958) is made by reformatting variable regions of the anti-FVIII BO2C11[21] onto cynoIgG1/kappa backbone[21] (Supplementary Table 3). Coding sequences of both heavy and light chains were codon-optimized at GeneArt (Thermo Fisher Scientific, Sunnyvale, CA, USA) and cloned into dual-expression UCOE-containing vector CET1019 HD-puro-SceI (Millipore, Billerica, MA), which was subsequently transfected into CHO-K1 cells (SAFC-Sigma, St. Louis, MO). CHO-K1 cell line stably expressing BO2C11-cynoIgG1 was generated, and protein was purified via Protein A affinity chromatography.

### Quantification of IgGs from cell culture supernatants.

Antibody expression levels were determined by protein A chromatography on an Agilent 1200 HPLC system (Agilent Technologies, Waldbronn, Germany). Samples were clarified by centrifugation for 5 min at 6700×g. A total of 900 µL of supernatant were mixed with 100 µL of 4.75 M NaCl and applied to a protein A column (POROS A/20 2.1 mm D × 30 mm L, Applied Biosystems, Foster City, CA, USA) previously equilibrated with 50 mM Na$_2$HPO$_4$/NaH$_2$PO$_4$ + 150 mM NaCl (pH 7.2) at a flow rate of 1.5 mL/min. After washing with 10 column volumes of the same buffer, IgGs were eluted with 50 mM Na$_2$HPO$_4$/NaH$_2$PO$_4$ + 150 mM NaCl (pH 3.1). The resulting peak was detected at OD 280 nm, integrated, and quantified using an IgG reference curve.

### Purification of plasma PC and production of species' APCs.

Generation of APCs in different species has been reported[15]. One liter of dog, rabbit, or monkey plasma was purchased as 20 × 50-mL frozen stocks with heparin included as anticoagulant (Bioreclamation, Inc., Westbury, NY). The purification method was a modified version of the method described previously[4,16]. Plasma was thawed at 4 °C and diluted 1:1 with 0.02 M Tris-HCl (pH 7.5), 1 U/mL heparin final, and 10 mM benzamidine HCl (final concentrations) at RT before loading onto a Q-Sepharose column for capturing PC and other vitamin K-dependent proteins. The column was washed with buffered 0.15 M NaCl, and PC was eluted with buffered 0.5 M NaCl. Eluents were recalcified with 10 mM CaCl$_2$ and 100 U/mL heparin, and then loaded onto HPC4-Affigel-10 affinity column (Bio-Rad, Hercules, CA, USA). The column was washed with CaCl$_2$-containing buffer and eluted with EDTA-containing buffer. Purified PC was dialyzed overnight into PBS buffer, flash-frozen, and stored at –80 °C as 0.5-ml aliquots. The purification yield was 1.75 mg from 1 L of dog plasma, 1.9 mg from 3 L of rabbit plasma, and 3.8 mg from 1 L of monkey plasma. The purified PC had 98% purity as determined by SDS-

PAGE and analytical SEC. PC was activated into APC using thrombin–agarose beads (Sigma-Aldrich).

**Antibody-binding kinetics analysis by SPR**. All experiments were performed with a Biacore T200 instrument at 25 °C with running buffer HBS-P + (10 mM HEPES, pH 7.4, 150 mM NaCl, and 0.05% surfactant P20) supplemented with 1 mg/mL BSA. Briefly, mAbs were captured by a mouse anti-human Fc antibody covalently immobilized to a CM5 sensor chip surface via amine coupling. hAPC was injected at 8 concentrations between 1.56 and 200 nM in 2-fold dilutions with running buffer (1.56, 3.125, 6.25, 12.5, 25, 50, 100, and 200 nM). The 25 nM concentration was run in duplicate. The surface of the capture antibody was regenerated after each antigen injection with 10 mM glycine–HCl, pH 2.0. Data obtained were double-referenced and fit to a 1:1 Langmuir binding model with a constant RI = 0.

**Enzyme-linked immunosorbent assay (ELISA)**. For antibody-binding specificity and species cross-reactivity studies, antigens (including hAPC, hPC, species' APC, species' PC, or PPACK-APC) were coated directly on Maxisorp 96-well plates (Thermo Fisher) at 100 ng/well in PBS/Ca$^{2+}$ buffer. After blocking the plate with 5% milk in PBS and washing with PBS-Tween-20, antibodies at various concentrations in PBS were added and incubated for 1 h at RT with shaking. The bound anti-APC IgG was detected by horseradish peroxidase (HRP)-conjugated anti-human Fc antibody followed by incubation with fluorogenic substrates (amplex red and H$_2$O$_2$) to produce fluorescence signals (relative fluorescence units, RFU) or with o-phenylenediamine dihydrochloride (OPD) to produce colorimetric signals at OD 490. All plates were read on a SpectraMax M2 microplate reader (Molecular Devices, Sunnyvale, CA, USA). For the antibody-selectivity study, human FIIa, FVIIa, FIXa, FXa, FXIa, and FXIIa were coated directly onto Maxisorp 96-well plates at 5–10 μg/mL. Anti-APC antibodies at 20 nM were tested for binding. As a positive control, an antibody specific for each antigen was used to demonstrate that the coating antigen was present. ELISA results are shown as means ± SDs from triplicate wells for each antigen concentration, and experiments were repeated more than three times.

**Crystallization, data collection, and structural determination**. Plasma-derived hAPC-Gla-domainless (APC-GD) was purchased from Enzyme Research Laboratories and characterized by ELISA to confirm that it could be recognized by both C25K23Fab and hu1573Fab. For APC-C25K23Fab complex formation, 0.9 mg of APC was mixed with 1.05 mg of C25K23Fab, and the reaction mixture was incubated at 4 °C for 5 h. The mixture was loaded onto a Superdex 200 10/300 GL column (GE Healthcare) equilibrated with HEPES-buffered saline (HBS, 10 mM HEPES, 150 mM NaCl, pH 7.2) 3 times to separate free APC or free Fab from the APC-C25K23Fab complex. Each fraction was collected and analyzed by SDS-PAGE under nonreducing and reducing conditions. The fractions containing the APC-C25K23Fab complex were pooled and concentrated to 16 mg/mL. For APC–hu1573Fab complex formation, 0.25 mg of APC was mixed with 0.25 mg of hu1573Fab, and the reaction mixture was incubated overnight at 4 °C. The mixture was loaded onto a Superdex 200 10/300 GL column (GE Healthcare) equilibrated with HBS buffer (10 mM HEPES, 150 mM NaCl, pH 7.2) to separate free APC or free Fab from the APC–hu1573Fab complex. Each fraction was collected and analyzed by SDS-PAGE under nonreducing and reducing conditions. This process was repeated 2–4 times, and the fractions containing the APC–hu1573Fab complex were pooled and concentrated to 15 mg/mL.

Crystallization of APC–C25K23Fab complex or APC–hu1573Fab complex under different crystal growth conditions was performed to produce crystals suitable for structure determination. The crystallization instrument Gryphon (Art Robbins Instruments, Sunnyvale, CA, USA) and ten commercial high-throughput crystallization screening kits (Hampton Research, Aliso Viejo, CA, USA, and Molecular Dimensions, Altamonte Springs, FL, USA) were used, and several hits were identified. The 2 hits for APC-C25K23Fab complex were as follows: (1) 0.1% n-octyl-beta-D-glucoside, 0.1 M sodium citrate tribasic dehydrate pH 5.5, 22% (w/v) PEG 3350, and (2) 18% 2-propanol, 0.1 M sodium citrate tribasic dehydrate, pH 5.5, 20% (w/v) PEG 4000. The best hits for aPC–hu1573Fab complex were (1) 0.2 M sodium fluoride, 20% (w/v) PEG 3350, (2) 0.1 M MES, pH 6.0, 0.2 M zinc acetate, 10% (w/v) PEG 8000, and (3) 0.1 M cacodylate, pH 6.5, 0.2 M lithium sulfate, 30% (v/v) PEG 400.

Diffraction data of crystals for both complexes were collected at the BL17U beamline at the Shanghai Synchrotron Research Facility. Diffraction data were indexed, integrated, and scaled with the program HKL2000[42] (HKL Research, Inc., Charlottesville, VA, USA). The structure was determined by molecular replacement with PHASER[42] in the CCP4 suite[43] using the published APC (PDB code: 1AUT)[16] and Fab crystal structures as models. Iterative refinement of individual coordinates was performed with COOT[44] and PHENIX[45], and model adjustment yielded models with $R_{work}$ of 0.201 and $R_{free}$ of 0.241 at a resolution of 2.2 Å for APC–C25K23Fab complex and $R_{work}$ of 0.278 and $R_{free}$ of 0.302 at a resolution of 3.7 Å for APC–hu1573Fab complex. All structural figures were made with PyMOL (PyMOL Molecular Graphics System, Schrödinger, LLC, Mannheim, Germany). Data processing and refinement statistics are listed in Table 1.

**APC amidolytic activity assay**. To assess the effect of antibody binding on the amidolytic activity of hAPC, hAPC at 20 nM was combined at equal volumes with the mAb at different concentrations (500, 167, 50, 16.7, 5, 1.67, and 0.5 nM) in a 96-well dilution plate and preincubated for 20 min at RT. The APC/antibody mixture was then transferred to assay plates at 50 μL/well in duplicates, followed by 50 μL of substrate Spectrozyme PCa at 1 mM. Substrate hydrolysis was monitored at 37 °C using a Spectramax reader at 405 nm for chromogenic reading.

**FVa inactivation by APC**. The effect of antibodies on the APC activity toward the biological substrate FVa was measured by a FXa and a thrombin-generation assay using purified FVa. In this assay, 20 pM FVa was incubated with 20 pM APC in assay buffer (25 mM Tris-HCl, 130 mM NaCl, 2.7 mM KCl, 25 μM phospholipids (PC:PE:PS 40:40:20), 5 mM CaCl$_2$, and 1 mg/mL BSA) in the presence of an antibody at 0, 1.6, 4.1, 10.2, 25.6, 64, 160, 400, 1000, 2500, and 5000 nM. After incubation for 30 min at RT, 100 μL of prothrombinase mixture containing 0.4 nM FXa, 1 μM prothrombin, and 0.6 mM thrombin chromogenic substrate was added to the 100-μL incubation mixture. The kinetics of thrombin-mediated substrate hydrolysis was monitored at 405 nm at RT by using a plate reader.

**TGA in the presence of TM–TGA**. To assess the procoagulant activity of mAbs, a modified TGA, TM-dependent TGA (TM–TGA), was used. In this assay, soluble TM (sTM) at 10 and 20 nM was added to normal human plasma and normal cynomolgus monkey plasma, respectively, to substitute for the endothelial cell surface TM. Added sTM facilitated plasma PC activation to APC, thus reducing total thrombin generation. Plasma TGA in the presence of sTM was initiated by 1 pM TF. The concentration of sTM was chosen empirically as the concentration that essentially abolished thrombin generation as initiated by 1 pM TF. An 80-μL mixture of antibody, sTM, and plasma was applied to 20 μL of reconstituted platelet-poor plasma reagent or calibrator. Each reaction was initiated with 20 μL of FluCa (Fluo substrate + CaCl$_2$) and incubated for at least 60 min in the thrombinoscope. Plasma alone was used as the negative control. The normal value for the TGA in normal human plasma for ETP is 800 ± 100 nM min and peak height is 70 ± 10 nM. Congenital HemA human plasma has ETP = 100 ± 50 nM min, peak < 5 nM. The normal value for the TGA in normal primate plasma for ETP is 2000 ± 100 nM min and peak height is 230 ± 10 nM. Induced HemA primate (cyno monkey) plasma had ETP = 700 ± 200 nM min, peak height = 50 ± 10 nM.

**Cellular TGA**. To provide a physiological surface for PC activation during TGA, confluent EA.hy926 endothelial cells were used to provide an endogenous source for TM and EPCR. EA.hy926 endothelial cells (ATTC #CRL-2922) were grown in black/clear-bottom 96-well plates (Costar #3603) in Dulbecco's Modified Eagle Medium (DMEM, Gibco #10313-021) supplemented with 10% fetal bovine serum (Omega #35-016-CV) and 1× penicillin/streptomycin/glutamine (Gibco #10378-016). Upon reaching confluency, cells were washed with Hanks' buffered salt solution (HBBS, Gibco #14175-095) supplemented with 1.2 mM CaCl$_2$, 0.6 mM MgCl$_2$, and 0.1% endotoxin-free BSA (Calbiochem #126579). Subsequently, 50 μl of 50% normal pooled plasma (George King) containing 1.45 μM Corn Trypsin Inhibitor (Heamtech), 10 mM CaCl$_2$, and 0.4 mM Gly–Gly–Arg–AMC (Bachem) in Hepes-buffered saline (20 mM Hepes, 147 mM NaCl, 3 mM KCl, and 0.1% endotoxin-free BSA, pH 7.4) was added to each well, and fluorescence (ex 360/em 460) was determined over time in a Spectramax Gemini EM fluorescence plate reader (Molecular Devices) at 37 °C. Type I, type II, isotype control IgG, anti-TM (50 μg/ml clone RTM96, HyCult Biotechnology), or anti-EPCR (50 μg/ml clone rcr-252, kindly provided by Dr. Keni Fukudome, Saga Medical School, Japan) were added as indicated. Fluorescence generation curves were corrected for inner filter effects, substrate depletion, and converted to nanomolar thrombin using the thrombin calibrator (Synapse) as described previously[46]. The ETP, defined as the area under the curve, was determined as described previously using Graphpad Prism V7.03 (GraphPad, San Diego, CA).

**aPTT in plasma**. To test the effects of mAbs on clotting time of normal human or monkey plasma or hemophilia plasma, we developed a Protac-dependent aPTT (Protac-aPTT). Clotting times were measured using the Stago STA Compact Coagulation System (Diagnostica Stago, Parsippany, NJ, USA). In this assay, 50% pooled normal human plasma was preincubated with buffer (no APC) or 400 ng/mL (6.7 nM) of hAPC or Protac at 0.03–0.2 IU/mL with various concentrations of mAb (0–554 nM) for 15 min at RT. A 50-μL sample was then mixed with 50 μL of automated aPTT reagent (DADE Behring, Marburg, Germany) containing cephalin (phospholipids extracted from rabbit brain in ellagic acid), and the mixture was incubated at 37 °C for 240 s. After the addition of 50 μL of 25 mM CaCl$_2$ solution, the time to clot formation was recorded. The normal value for the aPTT clotting time in NHP is 52 s. The normal value for the aPTT clotting time in normal primate plasma is ~20 s.

**APC cytoprotection assay**. Human umbilical vein endothelial cells (HUVEC) from ATCC (Manassas, VA, USA, Catalog# CRL-1730) were maintained in ATCC-formulated F-12K Medium (Catalog# 30-2004) supplemented with 10% fetal bovine serum. Cells were seeded at 2 × 10$^6$ cells per mL 1 d before the experiment. Confluent monolayers of HUVECs were incubated with 20 nM APC in the

presence of mAb (0, 3, 30, and 300 nM). Apoptosis was induced by incubation with calf thymus histone (Sigma-Aldrich) at 2 μM for 4 h. HUVECs were detached from the plate by rinsing once with PBS, and incubation with Enzyme-free Cell Dissociation Buffer (Gibco, Catalog#13151014) for 10 min followed by collection of the cells using a cell scraper, and cell viability was assessed using dyes, propidium iodide, or 7-AAD, as detected by fluorescence-activated cell-sorting analysis. The results are shown as mean (SD) of 3 independent experiments ($P < 0.01$ [ANOVA]).

**SEAP–PAR1 cleavage assay**. A PAR1 cleavage reporter construct was made with an N-terminal secreted embryonic alkaline phosphatase (SEAP) domain, and stable HEK293 cells expressing SEAP–PAR1 and wt-EPCR were obtained as reported previously[20,47]. APC in Hanks' balanced salt solution supplemented with 1.3 mM CaCl₂, 0.6 mM MgCl₂, and 0.1% BSA was incubated at RT with confluent SEAP–PAR1/wt-EPCR cells in 96-well plates in the presence and absence of 100 nM mAbs. After 60 min, SEAP release was determined using 1-step p-nitro-phenyl phosphate (Pierce #37621). After correction for background activity in the absence of APC, values were expressed as the percentage of the total SEAP activity present on the cells.

**APC-mediated PAR1 cleavage on endothelial cells**. APC-mediated cleavage of PAR1 on endothelial cells was performed as described using the Arg46 cleavage-site-sensitive antibody ATAP2[47]. EA.hy.926 cells were grown in black clear-bottom 96-well plates (Costar #3603) until confluent. Cells were washed and incubated at 37 °C with APC (100 nM) in HMM2 buffer (Hanks' buffered salt solution (HBBS, Gibco #14175-095) supplemented with 1.2 mM CaCl₂, 0.6 mM MgCl₂, and 0.1% endotoxin-free BSA (Calbiochem #126579)) for 3 h in the presence of 400 nM type I, type II, or control IgG. After a quick wash with ice-cold HMM2, cells were fixed in methanol-free 4% paraformaldehyde (Pierce #28908) for 20 min at RT, after which the plates were blocked in Odyssey blocking buffer (LI-COR #927-40000) supplemented with 3% BSA overnight. Subsequently, the plates were incubated with mouse anti-PAR1 antibody ATAP2 (10 μg/mL, kindly provided by Dr. L. Brass, University of Pennsylvania, Philadelphia, PA) in Odyssey blocking buffer, followed by 1/1000 biotinylated goat anti-mouse and 1/4000 IRDye 800CW streptavidin (LI-COR) with 1/10,000 Draq5 (Biostatus). PAR1 on the cell surface was detected using the In Cell Western module of Odyssey Imager with Image Studio Software Version 2.0. Fluorescence signals were corrected for cell number and background staining, and signals were normalized to buffer-only controls.

**Endothelial barrier assay**. Endothelial cell permeability was measured in real time using the iCelligence system (ACEA, San Diego, CA) that determines changes in TER by electric cell-substrate impedance sensing (ECIS). Briefly, EA.hy.926 endo-thelial cells were grown for 2 days to confluence in the 8-well culture dish containing gold-film electrodes. DMEM media (Invitrogen) containing 10% fetal calf serum was replaced with serum-free media containing 0.1% BSA, 2 h before addition of APC (50 nM) in the presence and absence of type I, type II, or control IgG. Permeability of confluent monolayers after treatment with APC for 4 h was determined upon addition of thrombin (2 nM). All comparisons of the cell index were made using normalized resistance. The results were expressed as percentage of maximal barrier disruption induced by thrombin in the absence of APC, or as percentage of maximal barrier protection by APC in the absence of mAbs. Wells were randomly assigned for experimental treatments.

**Half-life of APC in plasma**. Half-life of APC in the presence of type II mAb was determined as described[48] by measuring the residual amidolytic activity of exo-genously added APC in plasma over time. APC (70 nM) in Hepes-buffered saline (20 mM Hepes, 147 mM NaCl, 3 mM KCl, and 0.1% endotoxin-free BSA, pH 7.4) was added to 90% (vol/vol) citrated pooled human normal plasma (George King Biomedicals) in the presence and absence of 700 nM type II or control mAb and incubated at 37 °C. At selected times between 0 and 60 min, the residual APC amidolytic activity was determined by quenching 20 μl of the plasma sample in 80 μl of ice-cold tris-buffered saline, and immediately adding 50 μl of Pefachrom PCa (1 mM, Pentapharm #089-02) and monitoring the change in absorbance at 405 nm every 15 s for 5 min on a VersaMax (Molecular Devices) microplate reader. The amidolytic activity of APC at 0 min was set at 100%. The half-life ($t_{1/2}$) in plasma was determined by the exponential decay equation model by using GraphPad Prism.

**Testing type I and type II mAbs in normal monkeys**. All study protocols in this paper were approved by the Bayer HealthCare's and Prisys Biotechnologies' Institutional Animal Care and Use Committees, and were conducted in accordance with the Guide for the Care and Use of Laboratory Animals.

For pharmacokinetic studies, cynomolgus monkeys weighing 2–4 kg were used for the single-dose IV PK studies. Three cynomolgus monkeys were dosed with each mAb solution at 3 mg/kg over 2- to 5-min infusions. Blood samples were obtained before and immediately after dose administration, and at 5 min, 15 min, 30 min, 1 h, 4 h, 8 h, and 24 h after dosing on day 1. Additional samples were collected on days 3, 7, 8, 14, 16, and 21.

All blood samples were collected in 28-G butterfly needle syringes with prefilled anticoagulant (3.8% sodium citrate) at a 9:1 ratio (9 volumes of blood to 1 volume of 3.8% sodium citrate). Plasma was prepared by centrifugation of blood at 9000×$g$ for 10 min and stored at −80 °C until analysis.

Concentrations of anti-APC mAbs in monkey plasma samples were determined using both hIgG-capture ELISA (total mAb) and APC-capture ELISA (free mAb) with an estimated LLOQ of 78 μg/L mAb. Free and total concentrations of mAbs measured by 2 ELISA in monkey plasma were comparable. PK profiles are presented as concentration–time plots.

For toxicological studies, normal cynomolgus monkeys (male) in 4 groups were treated weekly with 0, 10, 30, and 100 mg/kg of type I mAb (TPP-2312/BAY1316786) or with 0, 5, 20, and 60 mg/kg type II mAb (TPP-4885/BAY1896502) by intravenous administration (2 monkeys per dose group) in a 15-day pilot study. They were planned to be sacrificed 24 h after the last dose. Clinical observations and food consumption were monitored daily. Coagulation markers (PT, aPTT, and fibrinogen), as well as biomarkers like PC and FVIII level, were measured in blood samples from the treated animals.

**Development of an acute bleeding model in HemA monkeys**. Monkeys were anesthetized with 5 mg/kg ketamine, and then maintained under inhalation anesthesia with 1% isoflurane at 0.8–1.5 liter/min flow rate of oxygen. Hemophilia A was induced in cynomolgus monkeys by the administration of 2.8 mg/kg poly-clonal sheep anti-human FVIII antibodies (Haemtologic Technologies, Essex Junction, VT) intravenously into the right arm ($T = -210$ min). At the site where the diameter of the tail is ~7 mm, a 2-cm section of the right tail vein was exposed without injuring the muscles or nerves. Monkeys were treated at $T = -15$ min with NovoSeven or anti-APC mAb. The vein was transected at $T = 0$ min, and the bleeding time was recorded. Bleeding time recording was stopped once no blood dripped into the centrifuge tube after a 30-s interval. At $T = 15$, 30, and 45 min, the clot at the transection site was disrupted by swabbing with a wet cotton tip applicator, and a new bleeding time was measured. The cumulative bleeding time from each clot disruption was combined and reported as the total bleeding time.

**Testing of type II mAb prophylactic efficacy in HemA monkeys**. For in vivo model development[49], cynomolgus monkeys were rendered hemophilic by the administration of 0.1 mg/kg monoclonal anti-FVIII BO2C11-cynoIgG1 intrave-nously into the right arm ($t = 0$). Bleeding was induced ($t = 2$ h) in the monkeys by inserting an 18-G needle 1-cm deep into the muscles at 16 sites (4 sites in each thigh, 2 sites in each upper arm, and 2 sites in each forearm). Anesthesia was maintained until the completion of this injury procedure. Blood samples (~2 mL) were collected from the forearm vein into ethylenediaminetetraacetic acid (EDTA)-K2 and sodium citrate tubes separately before type II mAb BAY 1896052 administration on day 7 and at various time points after anti-FVIII administration (1.5 h, 5.5 h, days 1–3). The blood samples collected into sodium citrate tubes at the designated time points were kept at RT, until centrifugation at 2300×$g$ at 22 °C for 10 min. Complete blood count and aPTT as well as hemoglobin measurements were performed. For prophylactic efficacy studies, cynomolgus monkeys were treated with 3, 10, or 30 mg/kg BAY 1896052 ($N = 3$ per dose group) or with saline ($N = 6$) by intravenous administration 7 days before induction of hemophilia A by 0.1 mg/kg of BO2C11-cynoIgG1 ($t = 0$). Blood samples were collected for hemo-globin assays using Sysmex XT 2000i Automated Hematology Analyzer (Systemex Corporation) from animals 72 h after BO2C11 administration.

**Reporting summary**. Further information on research design is available in the Nature Research Reporting Summary linked to this article.

## Data availability
The data that support the findings of this study are available from the corresponding authors upon reasonable request. Any Supplementary Information and Source Data files are available in the online version of the paper. The atomic coordinates and structure factors of the hAPC-C25K23Fab (type I) and hAPC-h1573Fab (type II) complex structures have been deposited in the Protein Data Bank (PDB) under accession codes 6M3B and 6M3C.

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

## Acknowledgements

The authors would like to thank Dr. Chuck Esmon (OMRF) for giving us the HAPC1573 hybridoma cell line to make this study possible. We also thank Yan Su, Kathy Tran, Bing Liu, Wei Xia, Warren Manzana, Andrea Eicker, Ruth Winter, Liang Tang, Yaoqi Chen, Thomas Thompson, Jinger Xie, Ming Bao, Faye Wu, Jun Shen, Karin J. Regnstrom, and Manfred Hoffmann for technical support. Editorial assistance was provided by Complete Healthcare Communications, LLC (West Chester, PA, USA) and was fully funded by Bayer.

## Author contributions

X.Y.Z., A.W., D.W., X.W., M.B., J-Y.K., J.M.G., P.E., T.M., Y.W., S.Y., D.S., S.Y., J-M.G., N.S., K.G., C.R-F., A.F., L.M., and A.H. designed and performed the experiments and analyzed the data; X-Y.Z., U.E., X.W., L.L., S.Y., C.P., D.S., N.S., T.H., L.M., and V.L. contributed to writing the paper; K.Mc.L., X.J., V.E., D.S., Y.Z., Y.X., D.M., J.T., L.L., Y.X., and C.M. performed the experiments.

## Competing interests

The authors declare the following competing interests: X.Y.Z., A.W., L.L., U.E., T.M., D.M., J.M.G., P.E., J.T., K.Mc.L., X.J., V.E., Y.X., A.H., V.L., N.S., Y.W.g, K.G., C.R-F., and A.F. are employees of Bayer. J.-M.G., M.B., J-Y.K., C.P., S.Y., D.S., Y.Z., S.Y., C.M., D.S., and T.H. were employees of Bayer at the time of the study. X.W., D.W., L.L., and L.M. received research funding from Bayer. X.W. is listed as a co-inventor of APC antibody patents filed by Bayer.

**Additional information**

