## [Peer Review File · Nature Communications]

Reviewers' Comments:

Reviewer #1:

Remarks to the Author:

The authors have adequately addressed all of my prior concerns. The paper is much improved and will be a valuable contribution to the field.

Reviewer #2:

Remarks to the Author:

The manuscript is very much improved in the current form, especially due to the inclusion of several of key data not in the original submission. There are some outstanding issues that should be addressed.

- 1) Reduction of PC level with type 1. In order to evaluate the difference between an active site inhibitor (type 1) and an exosite (near active site, type 2) binder in vivo, it is critical that evidence of binding to other proteins is absent. In this version of the manuscript the authors mention that the PC levels are reduced by 50% with type 1 and astoundingly that fVIII levels are massively reduced as well. This indicates that the type 1 antibody is not specific for APC, and one cannot rule out that other activities are contributing to the adverse events when given at high doses to monkeys. Why was this data left out of the original manuscript? Importantly, now that it is included one must acknowledge the possibility that the difference in tolerability of type 1 and 2 antibodies has nothing to do with APC signalling.
- 2) The fact that type 1 reduces PC activation by thrombin-TM is also troubling and suggests an extra activity. This is given only as a 'reviewer figure', but should be included in the manuscript. Again, this should temper the contention that the type 1 demonstrates that active site inhibition the only cause of the adverse in vivo effects observed.
- 3) At the end of the abstract there is a statement that 'only type II ...normalized hemostasis in monkeys'. Since only type 2 was tested, this is misleading.
- 4) The sentence at the end of page 9 is nonsensical: 'Whether the prolonged t1/2...'
- 5) The affinity of the antibodies to human APC and PC were tested by biocore and ELISA, but for NHP, it was only evaluated by ELISA and only for APC. One would think that the 50% reduction in PC levels in NHP would have prompted a thorough investigation into the binding of type 1 to cyno PC. These data need to be presented as they massively affect the interpretation of the cause of the adverse effects.
- 6) The reduction in PC levels by the same extent (50%) for 10, 30 and 100mg/kg type 1 is not commented on. Could this be the result of a saturation of the receptors used to take up the PC-Mab complex?
- 7) The most disturbing revelation in this new version is the catastrophic reduction in fVIII levels when dosed with type 1. Disturbing because it wasn't mentioned before, and because it demonstrates at least one important cross-reactant. Was binding to fVIII or vWF not investigated? Why is this not discussed in more detail? How is it possible to suggest that the adverse events like leaky vessels and hematomas are solely caused by the inhibition of APC signalling?!
- 8) The type 2 antibody clearly works in both hemostasis models, however, I am unclear as to why one is called acute and the other prophylactic. Indeed, I have no idea why the antibody was given 7 days

before the induction of haemophilia and challenge with needle sticks. This alone doesn't make it prophylactic. What would the difference have been if the antibody was given just after challenge? Was this tested?

Reviewer #3:

Remarks to the Author:

The authors have satisfactorily addressed this reviewer's comments. The inclusion of a more thorough evaluation of the impact of each antibody on APC cytoprotective activity in vitro and in the haemophilia A monkey model significantly strengthens the conclusions of the study.

Notably, the type II antibody does significantly attenuate EPCR-dependent APC proteolysis of PAR1 in transfected cells (Fig. 4b) and seems also to limit APC protection of the endothelial barrier at higher Ab concentrations (Fig. 4e), albeit not to the same extent as the type I antibody. Therefore, it seems possible that the type II antibody may impact APC cytoprotective activity when it is required in vivo. Experiments to assess this, are, however, beyond the scope of the existing study.

Manuscript # NCOMMS-19-1755118-T by Zhao *et al.*
Targeted Inhibition of Activated Protein C by a Non-Active-Site Inhibitory Antibody to Treat Hemophilia

Responses to Reviewers' Comments

We thank the Reviewers for their positive assessment of our manuscript and for their additional comments and suggestions. Our responses are noted underneath each of the Reviewers' comments.

Reviewers' comments:

Reviewer #1 - reviewer #2 at Nature Medicine (Remarks to the Author):

The authors have adequately addressed all of my prior concerns. The paper is much improved and will be a valuable contribution to the field. *Thank you.*

Reviewer #2 - reviewer #1 at Nature Medicine (Remarks to the Author):

The manuscript is very much improved in the current form, especially due to the inclusion of several of key data not in the original submission. *Thank you.*

There are some outstanding issues that should be addressed.

1) Reduction of PC level with type 1. In order to evaluate the difference between an active site inhibitor (type 1) and an exosite (near active site, type 2) binder in vivo, it is critical that evidence of binding to other proteins is absent. In this version of the manuscript the authors mention that the PC levels are reduced by 50% with type 1 and astoundingly that fVIII levels are massively reduced as well. This indicates that the type 1 antibody is not specific for APC, and one cannot rule out that other activities are contributing to the adverse events when given at high doses to monkeys.

The type I mAb induces an acquired Protein C deficiency. The reduction of factor VIII levels in the monkeys dosed with the type I mAb is consistent with the development of prothrombotic, proinflammatory and consumptive coagulopathy-like symptoms that have been extensively documented in mice and men with a non-functional protein C pathway. For instance, numerous publications report that a complete inhibition of all APC activities in mice with inhibitory anti-mouse PC/APC antibodies greatly increase the susceptibility to thrombotic and inflammatory complications and result in increased mortality (Reviewers' Reference #1-6). Even in the absence of a triggering event, mice that express low levels ($\leq 3\%$) of protein C suffer from spontaneous death due to DIC-like symptoms including thrombosis and hemorrhage, leading to necrosis of the ear, face, paws and legs often accompanied by severe edematous swelling and even resulting in spontaneous amputation of the individual extremities or the entire legs and tails (Reviewers' Reference #7). The description of these complications in mice with low protein C levels is very similar to that of purpura fulminans developing in human patients with severe protein C deficiency, including type 2 [Reviewers' Reference #8-9 (note: 2 selected reviews representing many available primary references)]. It is possible that some of the reduction in the protein C levels are also due to consumption since it is well known in the field that protein C levels are decreased in human patients with DIC (Reviewers' Reference #10).

In addition, the reduction of protein C levels is also explained by the residual affinity of the type I mAb for protein C, not by a cross-reactivity of the mAb to another protein as suggested by the

reviewer. As shown in Fig 1e and described in the manuscript, the type I mAb at higher concentrations binds to protein C. This notion is also supported by previous Reviewer's figure 2 – now Supplementary Figure 1 – demonstrating that the type I mAb binding to protein C reduced the activation of protein C by thrombin-thrombomodulin. The binding profiles of type I mAb to human and monkey protein C were very similar (Reviewers' Figure 1 and also see response to comment 5) and thus the observed reduction of protein C levels in the monkeys dosed with the type I mAb is consistent with some binding of the mAb to protein C. Our results are also consistent with published literature (e.g., Reviewers' Reference #2) demonstrating that an inhibitory anti-mouse PC/APC antibody (SPC-54) reduced circulating protein C levels by 25-50% in mice.

Also note that control experiments confirmed that the type I mAb did not interfere with the antibodies used in the ELISA to determine the PC levels in monkey plasma samples (data not shown).

We disagree with the reviewer that the adverse events observed with the type I mAb would indicate an off-target effect or other activities contributing to the adverse events. In fact, the deleterious effects of the type I mAb are exactly as predicted based on the extensive documentation of severe complications in mice and men with a non-functional protein C pathway. It was this knowledge and expectation that formed the basis to develop the type II mAb. Notwithstanding, the demonstration of the adverse effects in monkeys with the type I mAb is an important reminder that prohemostatic strategies encompassing inhibition of all APC functions are likely to encounter adverse effects similar to those observed with the type I mAb.

Reviewers' Figure 1.

ELISA for BAY1316786 Type I antibody: Antigen proteins (hAPC, hPC, monkey APC, or monkey PC) were coated directly on a MaxiSorp 96-well plate at 100 ng/well using PBS/Ca²⁺ buffer at 4°C overnight. After blocking with 5% milk and washing with PBS-Tween-20, Type I mAb (BAY1316786) at 100, 10, or 1 nM was added for antigen-antibody binding for 1 hr at room temperature. The bound anti-APC IgG was detected by horse radish peroxidase (HRP)-

conjugated anti-human Fc antibody followed by incubation with a fluorogenic substrate (Amplex Red and H₂O₂).

Why was this data left out of the original manuscript?

Our focus in the original manuscript was primarily on the characterization and efficacy of the type II antibody in hemophilia as this antibody has potentially the most direct clinical impact. Guided by the interest and suggestions of the reviewers, a more complete description of the type I adverse in vivo effects in monkeys was subsequently included.

Importantly, now that it is included one must acknowledge the possibility that the difference in tolerability of type 1 and 2 antibodies has nothing to do with APC signalling.

We respectfully, but strongly, disagree with the reviewer's assertion that it has nothing to do with APC signaling. The adverse effects of the type I mAb are caused by the absence of all APC activities that include APC's signaling functions. The increased susceptibility to death due to prothrombotic, proinflammatory and DIC-like complications associated with a lack of all APC functions has been well documented in humans and mice as detailed above.

2) The fact that type 1 reduces PC activation by thrombin-TM is also troubling and suggests an extra activity. This is given only as a 'reviewer figure', but should be included in the manuscript. Again, this should temper the contention that the type 1 demonstrates that active site inhibition the only cause of the adverse in vivo effects observed.

As requested, Reviewer's figure 2 has been incorporated in the text and is now Supplementary Figure 1. As detailed under comment 1, the reduction of protein C activation is explained by the residual affinity of the type I mAb for protein C. An additional control experiment was performed demonstrating that the type I mAb did not reduce the activation of thrombin activatable fibrinolysis inhibitor (TAFI) by thrombin-thrombomodulin (Reviewer's Figure 2) which indicates that the reduction of protein C activation was not caused by an extra activity of the mAb as suggested by the reviewer.

Reviewers' Figure 2.

Effects of type I mAbs on TAFI activation. *The activation of TAFI was determined by qualitative Western blot under the same conditions as previously used for protein C activation (Supplementary Figure 1) to demonstrate that the effect of the type I mAb on protein C activation was not due to an extra activity of the antibody. (a) Western blot of TAFI activation samples (200 nM TAFI, 5 nM thrombin, 10 nM thrombomodulin, 1 μM mAb) using mouse monoclonal anti-TAFI (9H10) and donkey anti-mouse-IRDye800 (green channel) for detection. The 9H10 anti-TAFI antibody is specific for the TAFI zymogen but does not react with activated TAFI or the*

activation peptide. (b) Quantification of TAFI activation in the presence of no mAb or type I mAb based on the disappearance of the TAFI zymogen. This control experiment demonstrates that the type I mAb has no off-target effects in this assay.

3) At the end of the abstract there is a statement that ‘only type II ...normalized hemostasis in monkeys’. Since only type 2 was tested, this is misleading.

This sentence at the end of the introduction now reads (change underlined): “Both mAbs effectively increased thrombin generation in vitro by blocking the anticoagulant activity of APC, but only type II retained APC’s cytoprotective function and was safe to normalize hemostasis in monkeys with hemophilia.”

4) The sentence at the end of page 9 is nonsensical: ‘Whether the prolonged t1/2...’

This sentence has been deleted to advance acceptance of the manuscript for publication.

5) The affinity of the antibodies to human APC and PC were tested by biocore and ELISA, but for NHP, it was only evaluated by ELISA and only for APC. One would think that the 50% reduction in PC levels in NHP would have prompted a thorough investigation into the binding of type 1 to cyno PC. These data need to be presented as they massively affect the interpretation of the cause of the adverse effects.

The binding of type I antibody to monkey PC/APC was tested and was unremarkably similar to human PC/APC as expected (Data in Reviewers’ Figure 1) since human and monkey PC share a 96% overall amino acid sequence identity and a 97% sequence similarity of the catalytic domain. All of the Type I antibody-contacting residues in human and monkey APC are conserved with the exception of 2 homologous substitutions at D60E and S95T that maintain the properties of these residues. Thus, there is no evidence to suggest that differences between the binding of Type I to human versus cyno PC/APC massively affects the cause of the adverse effects.

6) The reduction in PC levels by the same extent (50%) for 10, 30 and 100mg/kg type 1 is not commented on. Could this be the result of a saturation of the receptors used to take up the PC-Mab complex?

Possibly? It could also reflect the poor affinity of the type I mAb for protein C, or consumption of protein C by the DIC-like symptoms induced by the total inhibition of all APC functions by the type I mAb.

7) The most disturbing revelation in this new version is the catastrophic reduction in fVIII levels when dosed with type 1. Disturbing because it wasn’t mentioned before,

The type I mAb induces an acquired Protein C deficiency. The reduction of factor VIII levels in the monkeys dosed with the type I mAb is consistent with the development of prothrombotic, proinflammatory and consumptive coagulopathy-like symptoms that have been extensively documented in mice and men with a non-functional protein C pathway (see comment 1 for more details).

As stated before, the type II antibody was deemed to have the most direct clinical impact in hemophilia and therefore our focus in the original manuscript was primarily on the characterization and demonstration of efficacy for this antibody, rather than on the demonstration of the severe deleterious effects of total APC inhibition by the type I mAb. Guided

by the interest and suggestions of the reviewers, a more complete description of the type I adverse in vivo effects in monkeys was included in the revised manuscript.

and because it demonstrates at least one important cross-reactant. Was binding to fVIII or vWF not investigated? Why is this not discussed in more detail?

As already discussed under point 1 and 7 above, the reduction of factor VIII levels is entirely consistent with development of DIC-like complications in mice and men with non-functional protein C/APC. It does not reasonably demonstrate a cross-reactant. We have performed the requested experiment to advance acceptance of the manuscript for publication. As expected, the type I mAb showed no detectable binding to factor VIII.

How is it possible to suggest that the adverse events like leaky vessels and hematomas are solely caused by the inhibition of APC signalling?!

The adverse effects of the type I mAb are caused by the absence of all APC activities that include APC's signaling functions. The Reviewer makes an interesting conjecture by subtracting the effects of the type I and type II mAbs. However, the necessary tools to specifically inhibit the endogenous cytoprotective activities of APC have not been developed yet.

8) The type 2 antibody clearly works in both hemostasis models, however, I am unclear as to why one is called acute and the other prophylactic. Indeed, I have no idea why the antibody was given 7 days before the induction of haemophilia and challenge with needle sticks. This alone doesn't make it prophylactic. What would the difference have been if the antibody was given just after challenge? Was this tested?

Acute and prophylactic refer to the efficacy testing of the type II mAb as reflected in the paragraph heading. The sentence on p10 was corrected to avoid confusion and now reads "To evaluate the acute therapeutic effect of type II, a severe acquired HemA monkey acute bleeding model was developed using a sheep anti-human FVIII polyclonal antibody to induce HemA followed by anti-APC mAb treatment and tail vein injury."

Prophylactic efficacy testing of the type II mAb stems from the desire in the hemophilia field for longer lasting treatment options, especially in inhibitor patients but also in non-inhibitor patients, that requires less frequent administration as is currently needed with FVIII or FIX replacement product (e.g., 2-4 per week). To capture this aspect, the efficacy of the type II mAb was tested at 3, 10 and 30 mg/kg when given as a single administration 7 days in advance.

Reviewer #3 - new reviewer in replacement of reviewer #3 at Nature Medicine (Remarks to the Author):

The authors have satisfactorily addressed this reviewer's comments. The inclusion of a more thorough evaluation of the impact of each antibody on APC cytoprotective activity in vitro and in the haemophilia A monkey model significantly strengthens the conclusions of the study.

Thank you.

Notably, the type II antibody does significantly attenuate EPCR-dependent APC proteolysis of PAR1 in transfected cells (Fig. 4b) and seems also to limit APC protection of the endothelial barrier at higher Ab concentrations (Fig. 4e), albeit not to the same extent as the type I antibody. Therefore, it seems possible that the type II antibody may impact APC cytoprotective activity

when it is required in vivo. Experiments to assess this, are, however, beyond the scope of the existing study.

We agree with the Reviewer's assertion. The following sentence was added to the discussion.

“Notwithstanding the demonstrated safety of type II in exploratory monkey studies, future studies are needed to determine potential impact of the type II antibody on APC cytoprotective activities in vivo during inflammatory and/or ischemic challenge.”

Reviewers' References:

1. Xu J, Ji Y, Zhang X, Drake M, Esmon CT. Endogenous activated protein C signaling is critical to protection of mice from lipopolysaccharide-induced septic shock. *J Thromb Haemost.* 2009 May; 7(5):851-6.
2. Burnier L, Fernández JA, Griffin JH. Antibody SPC-54 provides acute in vivo blockage of the murine protein C system. *Blood Cells Mol Dis.* 2013 Apr; 50(4):252-8.
3. Kager LM, Wiersinga WJ, Roelofs JJ, Meijers JC, Zeerleder SS, Esmon CT, van't Veer C, van der Poll T. Endogenous protein C has a protective role during Gram-negative pneumosepsis (melioidosis). *J Thromb Haemost.* 2013 Feb; 11(2):282-92.
4. Schouten M, van 't Veer C, Poulussen N, Meijers JC, Levi M, Esmon CT, van der Poll T. The cytoprotective effects of endogenous activated protein C reduce activation of coagulation during murine pneumococcal pneumonia and sepsis. *Thromb Res.* 2015 Mar; 135(3):537-43.
5. Lattenist L, Jansen MP, Teske G, Claessen N, Meijers JC, Rezaie AR, Esmon CT, Florquin S, Roelofs JJ. Activated protein C protects against renal ischaemia/reperfusion injury, independent of its anticoagulant properties. *Thromb Haemost.* 2016 Jul 4; 116(1):124-33.
6. Chesebro BB, Rahn P, Carles M, Esmon CT, Xu J, Brohi K, Frith D, Pittet JF, Cohen MJ. Increase in activated protein C mediates acute traumatic coagulopathy in mice. *Shock.* 2009 Dec; 32(6):659-65.
7. Lay AJ, Liang Z, Rosen ED, Castellino FJ. Mice with a severe deficiency in protein C display prothrombotic and proinflammatory phenotypes and compromised maternal reproductive capabilities. *J Clin Invest.* 2005 Jun; 115(6):1552-61.
8. Goldenberg NA, Manco-Johnson MJ. Protein C deficiency. *Haemophilia.* 2008 Nov; 14(6):1214-21.
9. Chalmers E, Cooper P, Forman K, Grimley C, Khair K, Minford A, Morgan M, Mumford AD. Purpura fulminans: recognition, diagnosis and management. *Arch Dis Child.* 2011 Nov; 96(11):1066-71.
10. Yan SB, Helterbrand JD, Hartman DL, Wright TJ, Bernard GR. Low levels of protein C are associated with poor outcome in severe sepsis. *Chest.* 2001 Sep; 120(3):915-22.

Reviewers' Comments:

Reviewer #2:

Remarks to the Author:

The manuscript is improved with the inclusion of further data. The interpretation of the data is willfully simplistic in this reviewer's opinion, but enough data are included for readers to decide for themselves.

Reviewer #4:

Remarks to the Author:

Overall this is a thorough study of two antibodies with different binding specificity to activated protein C (APC): a type I active-site binding mAb and a type II mAb binding to an exosite on APC (required for anticoagulant activity). These antibodies have potential utility in treating coagulation disorders.

Some small additions in the presentation would improve the readability of the manuscript.

The humanization protocol for Fabs is given in great detail for both the Type I and Type II Fabs. However, the data presentation could be improved if the amino acid sequences of the resulting Fabs were given, possibly in the Supplementary information.

No description is provided about how the full length antibodies were constructed. Exactly what TPP-2313 (BAY 1316786) and TPP-4885 (BAY 1896502) are is unclear. Presumably they are hIgG2 but that is really only mentioned parenthetically on line 455. What vectors were used for their construction? In addition references should be provided for the constant region sequences used to construct BO2C11-cyno IgG1. The reference for BO2C11 is for the original hIgG4 antibody

Manuscript # NCOMMS-19-1755118-A by Zhao *et al.*
Targeted Inhibition of Activated Protein C by a Non-Active-Site Inhibitory Antibody to Treat Hemophilia

Responses to Reviewers' Comments

We thank the Reviewers for their positive assessment of our manuscript and for their additional comments and suggestions. Our responses are noted underneath each of the Reviewers' comments.

REVIEWERS' COMMENTS:

Reviewer #2 (Remarks to the Author):

The manuscript is improved with the inclusion of further data. The interpretation of the data is willfully simplistic in this reviewer's opinion, but enough data are included for readers to decide for themselves.

Reviewer #4 (Remarks to the Author):

Overall this is a thorough study of two antibodies with different binding specificity to activated protein C (APC): a type I active-site binding mAb and a type II mAb binding to an exosite on APC (required for anticoagulant activity). These antibodies have potential utility in treating coagulation disorders. Thank you!

Some small additions in the presentation would improve the readability of the manuscript. Agreed!

The humanization protocol for Fabs is given in great detail for both the Type I and Type II Fabs. However, the data presentation could be improved if the amino acid sequences of the resulting Fabs were given, possibly in the Supplementary information. As suggested by Reviewer#4, we are providing a new figure (as Supplementary Figure 10) to show the amino acid sequences of the resulting Fabs and mAbs of type I and type II antibodies as well as BO2C11-cynoIgG1 anti-FVIII antibody.

No description is provided about how the full length antibodies were constructed. Exactly what TPP-2313 (BAY 1316786) and TPP-4885 (BAY 1896502) are is unclear. For clarity, we added the following sentence to the first paragraph in Results: Both type I mAb (TPP-2312 or BAY 1316786) and type II mAb (TPP-4885 or BAY 1896502) are hIgG2.

Presumably they are hIgG2 but that is really only mentioned parenthetically on line 455. What vectors were used for their construction? More info is added to Line 455: When converting Fabs to full-length antibodies, hIgG2 isotype was used to minimize antibody effector functions. The sequences of full-length antibodies were first generated *in silico* by attaching the variable region sequences from Fab to the entire constant region sequences (hIgG2 CH1-CH2-CH3). Genes encoding the antibody heavy and light chains (please see amino acid sequences of heavy and light chains in Supplementary Figure 10) were subcloned separately in expression vector pTT5 (National Research Council Canada) and cotransfected into HEK293-6E cells³⁰

In addition references should be provided for the constant region sequences used to construct BO2C11-cyno IgG1. The reference for BO2C11 is for the original hIgG4 antibody. Yes, the reference on cynoIgG1 is added as Reference#16b.